# Stochastic Steepest Descent with Acceleration for $\ell_p$-Smooth Non-Convex Optimization

## Abstract

In this work, we analyze stochastic $\ell_p$ steepest descent for non-convex problems. Specifically, for $p > 2$, we establish $\epsilon$-approximate stationarity (in expectation) with respect to the dual norm $\|\cdot\|_{p^*}^{p^*}$ at a rate of $O(\epsilon^{-4})$, thereby generalizing the previous guarantees for signSGD ($p = \infty$). In addition, inspired by techniques for the convex setting, we present a new *accelerated* $\ell_p$ descent method, called Stacey, based on interpolated primal-dual iterate sequences that are designed for non-Euclidean smooth optimization settings. We compare our algorithm against popular methods such as SGD, Adam, AdamW, and Lion on image classification and pretraining language modeling tasks, and our results demonstrate the potential for both faster convergence and achieving higher accuracy. We further evaluate our algorithm for different values of $p$ across various models and datasets, highlighting the importance and efficiency of non-Euclidean methods as compared to standard Euclidean-based approaches.[1]

## 1 Introduction

Stochastic first-order methods have proven essential for efficiently training modern deep learning models. In addition to the basic stochastic gradient descent (SGD) algorithm (Robbins & Monro, 1951)—along with its momentum-based variants (Nesterov, 1983; Polyak, 1964)—other methods, such as AdaGrad (Duchi et al., 2011), Adam (Kingma, 2014), and AdamW (Loshchilov & Hutter, 2019), incorporate second moment gradient information to provide per-coordinate scaling, and the use of these adaptive techniques has since become standard for optimizing deep neural networks.

A related approach involves updating the parameters based on the *sign* of the (stochastic) gradient (Balles et al., 2020; Bernstein et al., 2018; Riedmiller & Braun, 1992). For example, the Lion method (Chen et al., 2023)—discovered symbolically through a program search—combines the sign-based step with a certain momentum scheme (which differs from that of the Signum method (Bernstein et al., 2018)), and more recently, the Lion-$\mathcal{K}$ method (Chen et al., 2024) establishes a family of methods—for which Lion is a special case—defined in terms of a general convex function $\mathcal{K}(\cdot)$. These algorithms have been shown to be competitive with—and in some cases even outperform—popular adaptive methods, particularly for large language models.

**Guarantees for non-convex optimization.** Given the empirical success of sign-based methods, we may then naturally ask *why* they perform as well as they do.[2] Although globally optimizing non-convex problems is NP-hard in general, one may nevertheless instead consider the relaxed goal of reaching approximate stationary points—sometimes strengthened to that of finding approximate local minima (Agarwal et al., 2017; Carmon et al., 2018; Ge et al., 2015)—for both deterministic (Carmon et al., 2017) and stochastic (Ghadimi & Lan, 2013) first-order methods. However, crucial to these guarantees (and their limitations) are the assumptions we make, notable among them being that the function is *smooth*, and there additionally lies behind these notions of stationarity (and smoothness) *a particular choice of norm*.

---

[1]The code is included in the supplementary material and will be publicly available upon acceptance.

[2]Indeed, understanding the dynamics—not to mention issues of generalization—for deep learning optimization has been the subject of significant effort (e.g., (Allen-Zhu et al., 2019a;b; Arora et al., 2019a;b; Du et al., 2017; 2018; 2019; Gunasekar et al., 2018a;b; Jacot et al., 2018; Jin et al., 2017; Soudry et al., 2018; Ward et al., 2020; Wilson et al., 2017)), such that a complete accounting is beyond the scope of this paper.

For example, in the case of SGD, Ghadimi & Lan (2013) establish approximate stationarity guarantees of the form $\mathbb{E}[\|\nabla f(\hat{x})\|_2] \leq \epsilon$ (where $\|\cdot\|_2$ denotes the standard Euclidean norm) under a smoothness assumption similarly defined with respect to $\|\cdot\|_2$. On the other hand, Bernstein et al. (2018) show how signSGD—which we may also view as (unscaled) stochastic steepest descent w.r.t. $\|\cdot\|_\infty$—can guarantee that $\mathbb{E}[\|\nabla f(\hat{x})\|_1] \leq \epsilon$, under a particular $\ell_2$ majorization assumption (which, as we discuss further in Appendix B, implies *smoothness w.r.t.* $\|\cdot\|_\infty$ (Balles et al., 2020)).

**Stochastic $\ell_p$ descent.**    Taken together, these two examples—albeit from opposite ends of the (norm) spectrum—suggest a fundamental interplay between the (primal) norm that is the basis of the *steepest descent* iteration (paired with smoothness defined in terms of the same norm) and the (dual) norm used to measure approximate stationarity. Previous works, however, have focused on either the case of stochastic steepest descent w.r.t. $\|\cdot\|_p$ for $p = 2$ (SGD) or $p = \infty$ (signSGD), or else depend on unconventional noise assumptions (Carlson et al., 2015), thus leaving open the question—which we address in Section 3—of extending these results to all $2 < p < \infty$ under standard variance assumptions.

While at first glance this may appear to be a straightforward extension, in fact several technical challenges arise when generalizing the analysis under $\ell_p$ smoothness assumptions, among them the fact that the stochastic coordinate-wise scaled step is not an unbiased estimator of the (deterministic) steepest descent direction (as for $p = 2$), nor is the magnitude the same across all coordinates of each step (as for $p = \infty$).[3] Indeed, extensions of this sort, in terms of general $\ell_p$ norms, for minimizing the dual norm of the gradient have been addressed in the deterministic, convex setting (Diakonikolas & Guzmán, 2024) (as have related questions for minimizing the optimality gap (Guzmán & Nemirovski, 2015; Nemirovskii & Nesterov, 1985)), and so our results provide a natural counterpart for the stochastic, non-convex setting.

Even so, one may reasonably ask: *why should we ever be concerned with any $p$ other than 2 or $\infty$?*

**Problem geometry and acceleration.**    In fact, we believe a key observation here lies in determining the *appropriate geometry* for the problem at hand, most clearly reflected in not only the choice of norm used for measuring smoothness, *but also the magnitude of the smoothness parameter itself*.[4] (This is naturally to be balanced against the different dual norms—e.g., $\|\nabla f(\hat{x})\|_2$ for $p = 2$ vs. $\|\nabla f(\hat{x})\|_1$ for $p = \infty$—used to define approximate stationarity.) Unfortunately, it can be difficult to determine the precise smoothness parameters w.r.t. general $\ell_p$ norms (Balles et al., 2020); nevertheless, there is ample evidence (e.g., Adolphs et al. (2019); Becker et al. (1988); Cohen et al. (2021a); Ghorbani et al. (2019); Jiang et al. (2024); Li et al. (2020); Li & Zhang (2024); Papyan (2018))—including empirical results of our own, as we later present in Section 5—to suggest that a different choice of $p$ (outside of 2 or $\infty$) could allow for better adapting to the structure of certain (deep learning) objectives.

As a complement to this matter of defining (and parameterizing) smoothness, however, there arises a second lens through which we observe the potential for general $p$, *namely that of acceleration* (Allen-Zhu & Orecchia, 2017; Bai & Bullins, 2024; Nemirovskii & Nesterov, 1985; Nesterov, 1983; 2005). Though we provide a more thorough overview in Section 4, there is, in essence, a fundamental trade-off (for convex settings) between the rate of acceleration and the norm used to measure the initial distance to the optimal solution. Concretely, it is well known that, for convex $f(x)$ that is $L$-smooth *with respect to* $\|\cdot\|_2$, the classic accelerated gradient descent (AGD) method of Nesterov (1983) converges at the rate $f(x_T) - f(x^*) \leq O\left(\frac{L\|x_0 - x^*\|_2^2}{T^2}\right)$, and this rate is indeed tight (Nesterov,

---

[3]Although stochastic mirror descent (S-MD) can provide guarantees (in terms of minimizing optimality gap, e.g., (Bubeck et al., 2015), Theorem 6.1) under smoothness w.r.t. a general norm $\|\cdot\|$, we would note that doing so requires the mirror map to be strongly convex w.r.t. $\|\cdot\|$, which leads to certain basic difficulties in optimization theory when considering $\|\cdot\|_p$ for $p > 2$. (We refer the reader to, e.g., (Cohen et al., 2021b; Kelner et al., 2014; Sherman, 2017; Sidford & Tian, 2018), for additional details.) This point likewise suggests what is a key difference between *steepest* and *mirror* descent (as also reflected in our analysis versus that of S-MD), the exploration of which yields interesting consequences for appropriately accelerating in *non-Euclidean* settings (Allen-Zhu & Orecchia, 2017), as we discuss in Section 4.

[4]We may note that, for all $2 \leq \gamma \leq \delta$, being $L_\delta$-smooth w.r.t. $\|\cdot\|_\delta$ implies being $L_\delta$-smooth w.r.t. $\|\cdot\|_\gamma$, whereas being $L_\gamma$-smooth w.r.t. $\|\cdot\|_\gamma$ implies being $d^{\frac{2}{\gamma} - \frac{2}{\delta}} L_\gamma$-smooth w.r.t. $\|\cdot\|_\delta$.

2018; Nemirovskij & Yudin, 1983). Importantly, we emphasize the appearance here of $\|\cdot\|_2$ for both the measure of smoothness *as well as* the $\|x_0 - x^*\|_2^2$ term.

**Trade-offs for non-Euclidean acceleration.** Based on the discussion so far, it would then be only natural to ask whether the accelerated rates of AGD hold under general smoothness assumptions. Unfortunately, the standard analysis of AGD does not readily adapt to alternative notions of smoothness, as the design of the algorithm is, in a sense, *specific to Euclidean settings*; we refer the reader to the work of Allen-Zhu & Orecchia (2017) for further discussion of this basic incompatibility. Nevertheless, several works (Diakonikolas & Guzmán, 2024; Nemirovskii & Nesterov, 1985; Nesterov, 2005; Song et al., 2019)—including that of Allen-Zhu & Orecchia (2017)—have provided techniques for *accelerating in non-Euclidean settings*, whereby common among them is, roughly speaking, a certain type of primal-dual coupling/interpolation. In particular, the approach of Nemirovskii & Nesterov (1985), for convex $f(x)$ that is $L$-smooth *with respect to* $\|\cdot\|_p$, leads to guarantees of the form

$$f(x_T) - f(x^*) \leq O\left(\frac{L\|x_0 - x^*\|_p^2}{T^{\frac{p+2}{p}}}\right). \tag{1}$$

(See also, e.g., Theorem 2 in (Diakonikolas & Guzmán, 2024).) Moreover, these rates are similarly known to be tight (Guzmán & Nemirovski, 2015).

Looking closely at these convergence guarantees, we may first note that, for $p = 2$, the rate in equation 1 recovers that of Nesterov (1983). On the other hand, for $p \to \infty$, while $\|x_0 - x^*\|_p^2$ can, at best, be as small as $d^{\frac{2}{p}-1}\|x_0 - x^*\|_2^2$, we also have that $\lim_{p\to\infty} T^{-\frac{p+2}{p}} = T^{-1}$—*in which case the benefit of acceleration disappears altogether*—and in fact this (limiting) rate essentially matches that of *unaccelerated $\ell_\infty$ steepest descent* (Kelner et al., 2014).

Consequently, these observations reveal the opportunity afforded by (non-Euclidean) $\ell_p$-based accelerated methods in the form of this trade-off between the *dependence on the problem geometry* and the *rate of acceleration*. As a further illustration, if we consider, e.g., $p = 4$, there is a (potential) gain of up to a $d^{1/2}$ factor (resulting from the $\|\cdot\|_4^2$ term) compared to the standard Euclidean ($p = 2$) case, whereas the rate of acceleration would degrade from $T^{-2}$ to $T^{-3/2}$.

**Practical considerations.** We acknowledge, of course, that these results are for convex problems, whereas in this work we focus on the non-convex setting.[5] Nevertheless, we would argue there is a well-established pattern (Agarwal et al., 2019; Dozat, 2016; Gupta et al., 2018; Kingma, 2014; Liu et al., 2020; 2024; Reddi et al., 2018; Sutskever et al., 2013; Zeiler, 2012) of designing deep learning optimizers in a manner inspired by those analyzed for *convex* settings (Boyd & Vandenberghe, 2004; Bubeck et al., 2015; Duchi et al., 2011; Nemirovskij & Yudin, 1983; Nesterov, 1983; Polyak, 1964; Robbins & Monro, 1951), and so we also work from such a starting point—our own inspiration drawing from *non-Euclidean* methods—in developing our new accelerated algorithm STACEY (**St**ochastic **St**eepest Descent with **Acce**leration), which we discuss further in Section 4.

### 1.1 CONTRIBUTIONS AND PAPER OVERVIEW

As a whole, the aim of this work is to examine more carefully the opportunities for non-convex problems *whose geometry is amenable to $\ell_p$ norm-based algorithms*. To this end, we begin by addressing in Section 3 the question of reaching $\epsilon$-approximate stationarity under general $\ell_p$ smoothness assumptions, whereby we establish, for $2 < p < \infty$, convergence guarantees of the form $E[\|\nabla f(\hat{x})\|_{p^*}^{p^*}] \leq \epsilon$ after $O(\epsilon^{-4})$ iterations of the stochastic $\ell_p$ descent algorithm (where we let $p^* := \frac{p}{p-1}$). We then present, in Section 4, our algorithm STACEY, which provides for *accelerating* these (stochastic) $\ell_p$ descent methods, based on a primal-dual interpolation of gradient and mirror descent steps. Finally, we observe the promising empirical performance of STACEY in Section 5, as demonstrated via both synthetic examples and large-scale image classification and pretraining language modeling tasks.

---

[5]In fact, under standard assumptions on the non-convex function (i.e., smoothness and being bounded from below) and the stochastic gradient oracle (i.e., that it is provides an unbiased estimator of the gradient with bounded variance), known lower bounds establish that, without additional assumptions, *acceleration is not attainable in general* (Arjevani et al., 2023; Carmon et al., 2020; 2021).

---

**Algorithm 1** Stochastic $\ell_p$ Descent

    **input** $p, \eta, f, \theta_0$

1: **for** $t = 0$ **to** $T - 1$ **do**

2:     $\theta_{t+1} = \theta_t - \eta s\left(g(\theta_t)\right)$            $\triangleright s(x) = [s_1(x), \cdots, s_d(x)]^\top$ where $s_i(x) = \frac{x^{(i)}}{|x^{(i)}|^{\frac{p-2}{p-1}}}$

    **return** $\theta_T$

---

## 2 PRELIMINARIES AND ASSUMPTIONS

Throughout we let $\|\cdot\|$ and $\|\cdot\|_*$ denote a general norm and its dual, respectively. In addition, we specify $\|\cdot\|_p$ to denote the standard $\ell_p$ norm ($1 \le p \le \infty$) and $\|\cdot\|_{p^*} := \|\cdot\|_{p/(p-1)}$ to denote its dual norm. For symmetric $M \in \mathbb{R}^{d \times d}$ s.t. $M \succ 0$, we further let $\|\cdot\|_M$ denote the standard matrix norm, i.e., $\|x\|_M = \sqrt{x^\top M x}$ for $x \in R^d$. For a vector $v \in \mathbb{R}^d$, we use superscript, i.e., $v^{(i)}$ to denote the $i^{th}$ coordinate of $v$, and we let $\mathrm{diag}(v)$ denote the diagonal matrix such that $\mathrm{diag}(v)_{i,i} = v^{(i)}$. We use subscript, e.g., $\theta_t$, to denote a vector in the $t^{th}$ iteration.

It will be useful for our analysis to consider certain basic regularity assumptions, such as that of smoothness.

**Definition 1** (Smoothness). *We say a function* $f : \mathbb{R}^d \mapsto \mathbb{R}$ *is L-smooth w.r.t.* $\|\cdot\|$ *if, for all* $x, y \in \mathbb{R}^d$, $\|\nabla f(y) - \nabla f(x)\|_* \le \|y - x\|$.

Equivalently, we have the following.

**Assumption 1** (Smoothness in $\ell_p$ norm). *Let* $f : \mathbb{R}^d \mapsto \mathbb{R}$ *be L-smooth w.r.t.* $\|\cdot\|_p$ *for* $p \ge 2$. *Then, for all* $x, y \in \mathbb{R}^d$,

$$\left| f(y) - f(x) - \nabla f(x)^\top (y - x) \right| \le \frac{L}{2} \|y - x\|_p^2.$$

**Assumption 2** (Unbiased Estimate). *The stochastic gradient* $g(x)$ *is an unbiased estimate of the true gradient* $\nabla f(x)$. *That is,* $\mathbb{E}[g(x)] = \nabla f(x)$.

**Assumption 3** (Bounded Variance). *For some data* $\xi$, *the variance of each coordinate of the stochastic gradient is bounded, i.e.,* $\forall i \in [d]$, $\mathbb{E}[|g(x)^{(i)} - \nabla f(x)^{(i)}|^2] \le \sigma_i^2$.

**Corollary 1.** *By Assumption 3,* $\mathbb{E}[\|g(x) - \nabla f(x)\|_2^2] \le \sigma^2$ *where for* $\sigma := \|\vec{\sigma}\|_2$, $\vec{\sigma} = [\sigma_1, \cdots, \sigma_d]^\top$.

**Corollary 2.** *If the stochastic gradient is an n-sample mini-batch estimate, then* $\forall i \in [d]$, $\mathbb{E}[|g(x)^{(i)} - \nabla f(x)^{(i)}|^2] \le \frac{\sigma_i^2}{n}$.

**Assumption 4** (Bounded gradient). *For* $G > 0$, $p \ge 2$, *and* $p^*$ *where* $\frac{1}{p} + \frac{1}{p^*} = 1$, $\|g(x)\|_{p^*} \le G$.

**Corollary 3.** *By Assumption 4, we know that*

(a) $\|\nabla f(x)\|_{p^*} = \|\mathbb{E}[g(x)]\|_{p^*} \le \mathbb{E}[\|g(x)\|_{p^*}] \le G$ *with Jensen's inequality.*

(b) $\forall i \in [d]$, $\left|g(x)^{(i)}\right| \le G$ *and* $\left|\nabla f(x)^{(i)}\right| \le G$.

## 3 CONVERGENCE FOR STOCHASTIC $\ell_p$ DESCENT

In this section, we present the stochastic $\ell_p$ descent algorithm and analyze its convergence. As demonstrated in Algorithm 1, the update step takes the unscaled form [6] of its counterpart in the deterministic setting $\theta_{t+1}^{(i)} = \theta_t^{(i)} - \eta \|f(\theta_t)\|_{p^*}^{\frac{p-2}{p-1}} \frac{f(\theta_t)^{(i)}}{|f(\theta_t)^{(i)}|^{\frac{p-2}{p-1}}}$ (Bai & Bullins, 2024), which is derived from the closed form of $\theta_{t+1} = \arg\min_\theta \left\{ \langle \eta f(\theta_t), \theta - \theta_t \rangle + \frac{1}{2} \|\theta - \theta_t\|_p^2 \right\}$. When $p = \infty$, Algorithm 1 reduces exactly to signSGD (Bernstein et al., 2018).

For $p > 2$, we show in Theorem 1 that stochastic $\ell_p$ descent converges in expectation to an $\epsilon$-approximate stationary point with respect to the dual norm at a rate of $O(\epsilon^{-4})$, thereby generalizing

---

[6]This is in line with signSGD (Bernstein et al., 2018) compared to the scaled form in (Balles et al., 2020). In addition, we adopt the unscaled version for clearer convergence analysis and more practical implementation.

the previous guarantees for signSGD ($p = \infty$). In addition, we provide a proof sketch, deferring the complete proof to Appendix A.1. Curiously, as we will see, moving from the $\ell_2$ setting (or even from the $\ell_\infty$ setting) introduces certain technical considerations that need to be addressed non-trivially.

**Theorem 1** (Main). *Running Algorithm 1 on some (possibly non-convex) function $f$ that satisfies Assumptions 1 to 4 yields*

$$\mathbb{E}\left[\frac{1}{T}\sum_{t=0}^{T-1}\|\nabla f(\theta_t)\|_{p^*}^{p^*}\right] \leq \frac{f(\theta_0) - f(\theta^*)}{\eta T} + \frac{1}{T}\sum_{t=0}^{T-1}\frac{\frac{2p-1}{p-1}G^{\frac{1}{p-1}}\|\vec{\sigma}\|_1}{\sqrt{n_t}} + \frac{L\eta G^{\frac{2}{p-1}}}{2}$$

*where $n_t$ is the batch size in iteration $t$ and $L$, $\vec{\sigma}$, and $G$ are constants from Assumption 1, 3, 4. Further letting the batch size $n_t = T$, the number of gradient call is $N = T^2$ for $T$ iterations. With $\eta = \frac{1}{L^{\frac{1}{2}}G^{\frac{1}{p-1}}T^{\frac{1}{2}}}$ we have*

$$\mathbb{E}\left[\frac{1}{T}\sum_{t=0}^{T-1}\|\nabla f(\theta_t)\|_{p^*}^{p^*}\right] \leq \frac{1}{N^{\frac{1}{4}}}\left[L^{\frac{1}{2}}G^{\frac{1}{p-1}}\left(f(\theta_0) - f(\theta^*) + \frac{1}{2}\right) + \frac{2p-1}{p-1}G^{\frac{1}{p-1}}\|\vec{\sigma}\|_1\right],$$

*i.e., Algorithm 1 takes $N \in \mathcal{O}\left(\epsilon^{-4}\right)$ gradient queries to reach an $\epsilon$-approximate stationary point.*

*Proof Sketch.* Starting with Assumption 1 and the descent step in Algorithm 1,

$$f(\theta_{t+1}) \leq f(\theta_t) - \underbrace{\eta\langle\nabla f(\theta_t),\, s(\nabla f(\theta_t))\rangle}_{A} + \underbrace{\eta\langle\nabla f(\theta_t),\, s(\nabla f(\theta_t)) - s(g(\theta_t))\rangle}_{B} + \underbrace{\frac{L\eta^2}{2}\|s(g(\theta_t))\|_p^2}_{C},$$

where $A = \eta\|\nabla f(\theta_t)\|_{p^*}^{p^*}$. In conventional first-order analysis, the inner product term $B$ is supposed to cancel out after taking expectation. In contrast, the closed-form stochastic $\ell_p$ descent update is coordinate-wise re-scaled, which makes the descent step *biased*, that is, $\mathbb{E}[s(g(x))] \neq s(f(x))$. In the literature on biased gradient descent (Stich & Ajalloeian, 2020; Demidovich et al., 2023), the bias terms simply accumulate as constants and do not decay with the iterations. Thus this term requires novel techniques to guarantee convergence. Noticing that $s_i(x) = \frac{x^{(i)}}{|x^{(i)}|^{\frac{p-2}{p-1}}} = \text{sign}(x^{(i)})|x^{(i)}|^{\frac{1}{p-1}}$,

$$B = \eta\sum_{i=1}^{d}\nabla f(\theta_t)^{(i)}\left(\text{sign}\left(\nabla f(\theta_t)^{(i)}\right)|\nabla f(\theta_t)^{(i)}|^{\frac{1}{p-1}} - \text{sign}\left(g(\theta_t)^{(i)}\right)|g(\theta_t)^{(i)}|^{\frac{1}{p-1}}\right)$$

$$= \eta\sum_{i=1}^{d}\left|\nabla f(\theta_t)^{(i)}\right|\left(|\nabla f(\theta_t)^{(i)}|^{\frac{1}{p-1}} + |g(\theta_t)^{(i)}|^{\frac{1}{p-1}}\right)\mathbb{I}\left[\text{sign}\left(\nabla f(\theta_t)^{(i)}\right) \neq \text{sign}\left(g(\theta_t)^{(i)}\right)\right]$$

$$+ \eta\sum_{i=1}^{d}\left|\nabla f(\theta_t)^{(i)}\right|\left||\nabla f(\theta_t)^{(i)}|^{\frac{1}{p-1}} - |g(\theta_t)^{(i)}|^{\frac{1}{p-1}}\right|\mathbb{I}\left[\text{sign}\left(\nabla f(\theta_t)^{(i)}\right) = \text{sign}\left(g(\theta_t)^{(i)}\right)\right].$$

Denote the first term as $B_1$ and the second $B_2$. The $|\nabla f(\theta_t)^{(i)}|^{\frac{1}{p-1}} + |g(\theta_t)^{(i)}|^{\frac{1}{p-1}}$ term in $B_1$ can be bounded by $2G^{\frac{1}{p-1}}$ with Corollary 3, after which we take expectation, turning the indicator into a probability, and Lemma 2 in Appendix A.1 shows $\mathbb{E}\left[B_1\right] \leq \frac{2\eta G^{\frac{1}{p-1}}\|\vec{\sigma}\|_1}{\sqrt{n_t}}$ using Markov's inequality.

$B_2$ requires more sophisticated handling since we cannot push the expectation through due to the data dependence of the term $\left||\nabla f(\theta_t)^{(i)}|^{\frac{1}{p-1}} - |g(\theta_t)^{(i)}|^{\frac{1}{p-1}}\right|$, nor does $\mathbb{P}\left[\text{sign}\left(\nabla f(\theta_t)^{(i)}\right) = \text{sign}\left(g(\theta_t)^{(i)}\right)\right]$ give us much information. We instead take the zeroth-order Taylor expansion so that $\forall\, i \in [d],\, \exists\, \zeta^{(i)}$ between $\nabla f(\theta_t)^{(i)}$ and $g(\theta_t)^{(i)}$ such that

$$|\nabla f(\theta_t)^{(i)}|^{\frac{1}{p-1}} = |g(\theta_t)^{(i)}|^{\frac{1}{p-1}} + \frac{1}{p-1}\text{sign}(\zeta^{(i)})\left|\zeta^{(i)}\right|^{\frac{2-p}{p-1}}\left(\nabla f(\theta_t)^{(i)} - g(\theta_t)^{(i)}\right)$$

And $\left||\nabla f(\theta_t)^{(i)}|^{\frac{1}{p-1}} - |g(\theta_t)^{(i)}|^{\frac{1}{p-1}}\right| = \frac{1}{p-1}\text{sign}(\zeta^{(i)})\left|\zeta^{(i)}\right|^{\frac{2-p}{p-1}}\left(\nabla f(\theta_t)^{(i)} - g(\theta_t)^{(i)}\right)$. Furthermore, given $\text{sign}\left(\nabla f(\theta_t)^{(i)}\right) = \text{sign}\left(g(\theta_t)^{(i)}\right)$, it is either $\left|\nabla f(\theta_t)^{(i)}\right| \leq \left|\zeta^{(i)}\right| \leq \left|g(\theta_t)^{(i)}\right|$ or

---

**Algorithm 2** STACEY$_{(p,2)}$ Optimizer

---

    **input** $p, \beta_1, \beta_2, \alpha, \tau, \eta, \epsilon, \lambda, f$
    **initialize** $\theta_0, z_0, m_0 \leftarrow 0$
1: **while** $\theta_{t+1}$ not converged **do**
2:      $g_t \leftarrow g(\theta_t)$                           $\triangleright$ $g(\theta_t)$ s.t. $\mathbb{E}[g(\theta_t)] = \nabla f(\theta_t)$
3:      $c_{t+1} \leftarrow \beta_1 m_t + (1 - \beta_1) g_t$
4:      $y_{t+1} \leftarrow \theta_t - \eta_t s_\epsilon (c_{t+1})$     $\triangleright$ $s^\epsilon(x) = [s_1^\epsilon(x), \cdots, s_d^\epsilon(x)]^\top$ where $s_i^\epsilon(x) = \frac{x^{(i)}}{|x^{(i)}|^{\frac{p-2}{p-1}} + \epsilon}$
5:      $z_{t+1} = z_t - \alpha c_{t+1}$
6:      $\theta_{t+1} = \tau z_{t+1} + (1 - \tau) y_{t+1} - \eta_t \lambda \theta_t$
7:      $m_{t+1} = \beta_2 m_t + (1 - \beta_2) g_t$
    **return** $\theta_{t+1}$

---

$\left| \nabla f(\theta_t)^{(i)} \right| \geq \left| \zeta^{(i)} \right| \geq \left| g(\theta_t)^{(i)} \right|$. Appendix A.1 Lemma 3 shows that $\mathbb{E}[B_2] \leq \frac{\eta G^{\frac{1}{p-1}} \|\vec{\sigma}\|_1}{(p-1)\sqrt{n_t}}$ in either case.

Term $C$ is usually turned into mean-squared error that coincides with variance in an unbiased setting, which the bounded variance assumption can directly handle. This is not the case for our setting. It is worth noting that the analysis of signSGD (Bernstein et al., 2018), a special case of the $\ell_p$ setting with $p = \infty$, was able to push through due to its update being in the very form of the sign of the gradient, which is in itself bounded by the constant 1. Our update, in contrast, is much more complicated with the absolute value of the coordinates of the gradient in the denominator, which is only lower bounded 0, or some $\epsilon > 0$ at best. Therefore, we directly apply Assumption 4 and $C = \frac{L\eta^2}{2}\|\nabla f(\theta_t)\|_{p^*}^{\frac{2}{p-1}} \leq \frac{L\eta^2 G^{\frac{2}{p-1}}}{2}$. Moving term $A$ to the left hand side, telescoping through iteration 0 to $T - 1$, and dividing both sides by $\eta T$ completes the proof. $\qquad\square$

## 4    Accelerating Stochastic Steepest Descent

Building on the *unaccelerated* stochastic $\ell_p$ descent for non-convex settings, we present accelerated versions of the method through the interpolation of two sequences in primal and dual spaces. Indeed, this type of interpolation is the basis of the linear coupling framework (Allen-Zhu & Orecchia, 2017), wherein a steepest descent step is carefully coupled with a mirror descent step. Similar "coupling" can also be found in Nesterov's generalization of standard AGD to non-Euclidean settings (Nesterov, 2005) and recent acceleration for $\ell_p$ descent in the deterministic convex setting (Bai & Bullins, 2024). Inspired by these previous examples (and their successes, e.g., (Bullins, 2020; Jambulapati et al., 2019; Sherman, 2017; Sidford & Tian, 2018)), we introduce a practical acceleration scheme called STACEY, which is *specifically designed for non-Euclidean methods*. As presented in Algorithm 2, the algorithm takes the steepest descent step with respect to the $\ell_p$-norm in line 4 and then a gradient step in line 5. The update on the variable $\theta$ is an interpolation between the two, controlled by the parameter $\tau$. The algorithm generalizes linear coupling (Allen-Zhu & Orecchia, 2017) with non-Euclidean steepest descent while taking the mirror descent step with the distance generating function chosen as $\frac{1}{2}\|\cdot\|_2^2$. We further specify the name as STACEY$_{(p,2)}$ to clarify the norms in which the steepest descent and mirror descent steps are taken.

We wish to note that even though for smooth convex optimization, (deterministic) gradient descent can be accelerated to achieve a rate of $O(1/T^2)$, for stochastic first-order methods, however, it has been shown that a) in convex settings, SGD cannot improve upon the standard $O(1/\sqrt{T})$ rate when noise parameter $\sigma$ is large enough (Agarwal et al., 2009), and b) in first-order smooth *non-convex* settings, *SGD cannot be accelerated (in theory)* without additional assumptions (in terms of gradient norm minimization), due to known lower bounds (Arjevani et al., 2023). Nevertheless, standard practical implementations of SGD are frequently designed to introduce *some* notion of acceleration with momentum (e.g., (Bernstein et al., 2018; Sutskever et al., 2013)),[7] "pushing" the converging sequence further along the direction of previous gradients.

---

[7]Momentum coincides with Nesterov's acceleration in the deterministic convex setting, though this by no means makes them equivalent in stochastic non-convex settings.

**Algorithm 3** $\text{STACEY}_{(p,p)}$ Optimizer

> **input** $p, \beta_1, \beta_2, \alpha, \tau, \eta, \epsilon, \lambda, f$
> **initialize** $\theta_0, z_0, m_0 \leftarrow 0$

1: **while** $\theta_{t+1}$ not converged **do**
2:      $g_t \leftarrow g(\theta_t)$        $\triangleright g(\theta_t)$ s.t. $\mathbb{E}[g(\theta_t)] = \nabla f(\theta_t)$
3:      $c_{t+1} \leftarrow \beta_1 m_t + (1 - \beta_1) g_t$
4:      $y_{t+1} \leftarrow \theta_t - \eta_t s_\epsilon (c_{t+1})$    $\triangleright s^\epsilon(x) = [s_1^\epsilon(x), \cdots, s_d^\epsilon(x)]^\top$ where $s_i^\epsilon(x) = \dfrac{x^{(i)}}{\left|x^{(i)}\right|^{\frac{p-2}{p-1}} + \epsilon}$

5:      $z_{t+1}^{(i)} = \dfrac{\left|z_t^{(i)}\right|^{p-2} z_t^{(i)} - \alpha c_{t+1}^{(i)}}{\left|\left|z_t^{(i)}\right|^{p-2} z_t^{(i)} - \alpha c_{t+1}^{(i)}\right|^{\frac{p-2}{p-1}}}, \forall\, i \in [d]$

6:      $\theta_{t+1} = \tau z_{t+1} + (1 - \tau) y_{t+1} - \eta_t \lambda \theta_t$
7:      $m_{t+1} = \beta_2 m_t + (1 - \beta_2) g_t$
   **return** $\theta_{t+1}$

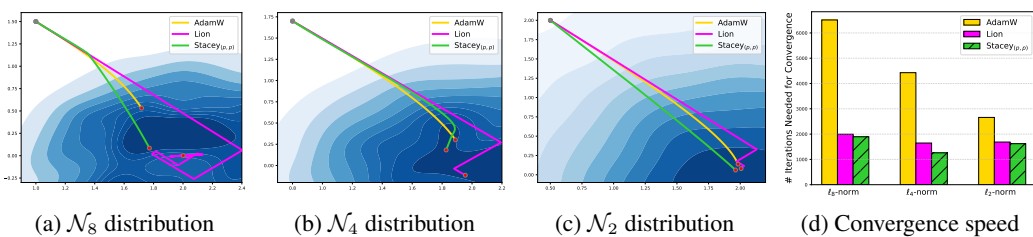

      (a) $\mathcal{N}_8$ distribution      (b) $\mathcal{N}_4$ distribution      (c) $\mathcal{N}_2$ distribution      (d) Convergence speed

Figure 1: Results on synthetic $p$-generalized Gaussian distributions. $\text{STACEY}$ optimizer is more stable on highly non-Euclidean distributions, and converges faster than AdamW and Lion.

In contrast, we take the view of acceleration not as a "pushing" (in the Euclidean sense), but rather as a (dynamic) interpolation of two iterate sequences: one acting from a (primal) steepest descent perspective (line 4 Algorithm 2), while the other functions in a dual capacity (line 5 Algorithm 2). An obvious distinction (pun intended) is momentum, as a separate functionality, can be applied on top of the acceleration scheme in $\text{STACEY}_{(p,2)}$, as demonstrated in lines 3 and 7 of Algorithm 2, for both the steepest descent and the gradient descent.

In the realm of non-Euclidean methods, we contrast our algorithm with Lion-$\mathcal{K}$ (Chen et al., 2024; Bernstein et al., 2018). While at first glance it may seem that these methods may simply be a rewriting of each other (based on the choice of parameters), a closer inspection on *the very first step* reveals that such is not the case:

$$\text{Lion-}\mathcal{K}: \theta_1 = -\eta \nabla \mathcal{K} \left((1 - \beta_1) g(\theta_0)\right),$$
$$\text{STACEY}_{(p,2)}: \theta_1 = -(1 - \tau)\eta s^\epsilon \left((1 - \beta_1) g(\theta_0)\right) - \tau \alpha (1 - \beta_1) g(\theta_0).$$

where $\mathcal{K}(\cdot) = \|\cdot\|_p$ and $s^\epsilon(\cdot)$ is defined in Algorithm 2. The key difference of $\text{STACEY}_{(p,2)}$ lies in the convex combination of a steepest descent step and a gradient descent step, whereas Lion-$\mathcal{K}$ is composed of only the steepest descent step. They only coincide when $\tau = 0$ for $\text{STACEY}_{(p,2)}$, i.e., completely getting rid of the "coupling", which then defeats the purpose of our acceleration. In addition, there is no choice of parameters for Lion-$\mathcal{K}$ to recover linear coupling. As a result, they are not iterate-equivalent, which further highlights the fundamental difference between "momentum" and "acceleration", a distinction which, crucially, does not appear in the case of standard (Euclidean) AGD, i.e., when both steepest and mirror descent steps are with respect to Euclidean norms.

Further inspired by the fact that $\text{STACEY}_{(p,2)}$ breaks the symmetry (in primal and dual trajectories) by coupling an $\ell_p$ steepest descent step with an $\ell_2$-based mirror descent step, we present the natural variant $\text{STACEY}_{(p,p)}$ (Algorithm 3), for which we group $\ell_p$ steepest descent with a mirror descent step having $\frac{1}{p}\|\cdot\|_p^p$ (whose $p^{th}$-order uniform convexity is useful for non-Euclidean acceleration (Song et al., 2019)) as its distance generating function. The closed-form mirror descent update is presented in line 5 of the algorithms.

Table 1: Image classification on CIFAR at the 50th, 100th, and 200th epochs. STACEY consistently outperforms other optimizers at all epochs, demonstrating both superior accuracy and faster convergence.

| Optimizer | Training NLL ↓ | | | Testing ACC (%) ↑ | | |
|---|---|---|---|---|---|---|
| | @50 epoch | @100 epoch | @200 epoch | @50 epoch | @100 epoch | @200 epoch |
| SGD w/ Nesterov | 0.0523 | 0.0342 | 0.0289 | 91.78 | 91.93 | 92.69 |
| Adam | 0.1303 | 0.0487 | 0.0229 | 90.03 | 90.63 | 91.58 |
| AdamW | 0.0620 | 0.0298 | 0.0170 | 89.99 | 91.39 | 91.89 |
| Lion | 0.0410 | 0.0199 | 0.0103 | 91.85 | 92.48 | 92.69 |
| STACEY$_{(p,p)}$ | 0.1438 | 0.0405 | 0.0006 | 88.95 | 91.50 | **94.05** |
| STACEY$_{(p,2)}$ | **0.0375** | **0.0104** | **0.0005** | **91.87** | **92.92** | 93.99 |

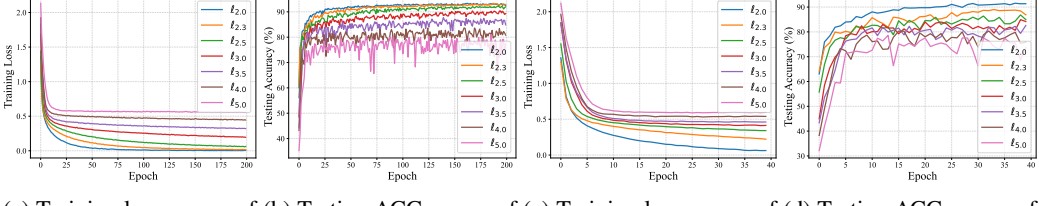

(a) Training loss curves of STACEY$_{(p,p)}$    (b) Testing ACC curves of STACEY$_{(p,p)}$    (c) Training loss curves of STACEY$_{(p,2)}$    (d) Testing ACC curves of STACEY$_{(p,2)}$

Figure 2: Learning curves of CIFAR classification with varying $\ell_p$-norm.

## 5 EXPERIMENTS

In this section, we present empirical evidence that the STACEY optimizer outperforms other optimizers in both convergence speed and accuracy. We evaluate STACEY's effectiveness on synthetic distributions (Section 5.1), image classification (Section 5.2), and LLM pretraining (Section 5.3). The hyperparameter choices are summarized in Appendix D.

In all experiments, we underscore the efficiency of the STACEY optimizer by comparing it against other optimizers as baselines including SGD (with Nesterov's momentum) (Nesterov, 1983), Adam (Kingma, 2014), AdamW (Loshchilov & Hutter, 2019), and Lion (Chen et al., 2023). For synthetic distribution estimation, we demonstrate that STACEY outperforms Lion and AdamW in convergence speed on generated $\ell_p$ Gaussian datasets.

In real-world large datasets, such as training from scratch on ImageNet (Deng et al., 2009) and LLM (LLaMA (Touvron et al., 2023)) pretraining on C4, we further demonstrate the necessity of utilizing different $\ell_p$-norms for specific tasks. For example, in the CIFAR image classification, an $\ell_p$-norm for $p$ close to 2 delivers the best performance (Section 5.2), consistent with the effectiveness of Euclidean-based optimizers. In contrast, a $\ell_p$-norm with $p$ around 3 proves more effective in LLM pertaining (Section 5.3). These results highlight the importance of developing non-Euclidean optimizers and adjusting the choice of $\ell_p$-norm to enhance performance across different tasks.

### 5.1 ESTIMATING SYNTHETIC DISTRIBUTIONS

STACEY optimizer is designed for generalized $\ell_p$-norm optimization with $p \geq 2$. Following D'Angelo & Fortuin (2021); Li & Zhang (2024), we visualize the trajectory of optimizers when estimating synthetic distributions in Fig. 1, to demonstrate STACEY's faster convergence compared to other optimizers on $p$-generalized Gaussian distributions (Subbotin, 1923; Kalke & Richter, 2013). The synthetic distributions $\mathcal{N}_p(\boldsymbol{\mu})$ marginally follow the $p$-generalized Gaussian distribution whose *probability density function* (PDF) is given by $p\left(\mathbf{x}^{(i)}\right) = \frac{p^{1-1/p}}{2\Gamma(1/p)} \exp\left\{-\left|\boldsymbol{x}^{(i)} - \boldsymbol{\mu}^{(i)}\right|^p /p\right\}$, and thus the PDF of $\mathcal{N}_p(\boldsymbol{\mu})$ is

$$p\left(\boldsymbol{x}\right) = \prod_{i=1}^d p\left(\mathbf{x}^{(i)}\right) \propto \exp\left\{-\sum_{i=1}^d \frac{\left|\boldsymbol{x}^{(i)} - \boldsymbol{\mu}^{(i)}\right|^p}{p}\right\} = \exp\left\{-\frac{\|\boldsymbol{x} - \boldsymbol{\mu}\|_p^p}{p}\right\}.$$

Table 2: Image classification on ImageNet at the 20th, 50th, and 90th epochs. STACEY consistently outperforms other optimizers at all epochs, demonstrating both superior accuracy and faster convergence.

| Optimizer | Training NLL ↓ | | | Testing Top-1 ACC (%) ↑ | | |
|---|---|---|---|---|---|---|
| | @20 epoch | @50 epoch | @90 epoch | @20 epoch | @50 epoch | @90 epoch |
| SGD | 3.9729 | 2.4376 | 1.9257 | 21.05 | 45.94 | 63.17 |
| STACEY$_{(p,p)}$ | **1.9371** | **1.2064** | **0.9902** | **60.84** | **68.23** | **69.88** |
| STACEY$_{(p,2)}$ | 3.3706 | 2.5149 | 2.1975 | 32.16 | 49.39 | 57.33 |

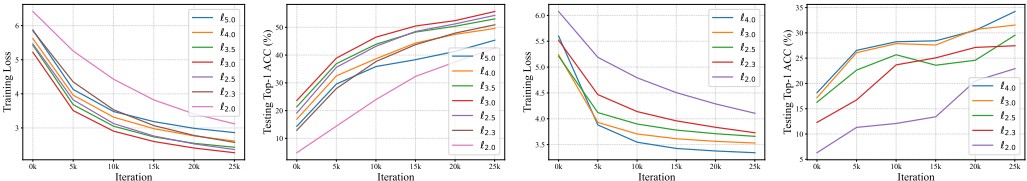

(a) Training loss curves of STACEY$_{(p,p)}$ (b) Testing ACC curves of STACEY$_{(p,p)}$ (c) Training loss curves of STACEY$_{(p,2)}$ (d) Testing ACC curves of STACEY$_{(p,2)}$

Figure 3: Learning curves of ImageNet classification at the first 6 epochs with varying $\ell_p$-norm.

We sample synthetic datasets from $\mathcal{N}_p([2,0]^T)$ distributions with varying $p$ values, where larger $p$ typically yields more complex non-Euclidean optimization problems. For each optimizer, we set their learning rates to be $10^{-3}$ and plot 5000-iteration trajectories. Results show that STACEY maintains stable convergence even with larger $p$ values. In contrast, AdamW (Loshchilov & Hutter, 2019) converges more slowly, and Lion (Chen et al., 2023) exhibits significant fluctuations.

Fig. 1d compares the average convergence rates of different optimizers. We initialize points from a standard Gaussian distribution and repeat each experiment 100 times. Results show that STACEY converges faster than AdamW and Lion, especially on the highly non-Euclidean $\mathcal{N}_8$ distribution.

## 5.2 IMAGE CLASSIFICATION

We demonstrate improved accuracy and faster convergence of the STACEY optimizer across image classification tasks of varying scales, consistent with our algorithm's design for acceleration.

**Training from scratch on CIFAR.** We train ResNet18 (He et al., 2016) on the CIFAR dataset (Krizhevsky, 2009) for 200 epochs, with the results presented in Table 1. We report training NLL and testing accuracy at the 50th, 100th, and 200th epochs. The proposed STACEY optimizer consistently outperforms all compared optimizers. As shown in Fig. 2, a $p$-norm of 2 yields the best performance for the CIFAR dataset when using the ResNet18 architecture.

**Training from scratch on ImageNet.** We train ResNet50 (He et al., 2016) with a batch size 256[8] on ImageNet (Deng et al., 2009) for 90 epochs. The learning rate schedule is cosine with 10K steps warm up, and the momentum is saved as `bfloat16` to reduce the memory footprint. The learning curves are shown in Table 2.

## 5.3 PRETRAINING LARGE LANGUAGE MODELS (LLMS)

We pretrain LLaMA 100M (Touvron et al., 2023) on the C4 dataset[9] using various optimizers. The learning curves, presented in Fig.4, show that the STACEY optimizer outperforms the alternatives. Additionally, Fig.5 indicates that a $p$-norm of 3 yields the best performance, which contrasts with the optimal $p = 2$ observed in the CIFAR image classification tasks discussed in Section 5.2.

---

[8]Our batch size 256 is significantly smaller than Lion's (Chen et al., 2024) batch size 1024.
[9]https://huggingface.co/datasets/allenai/c4

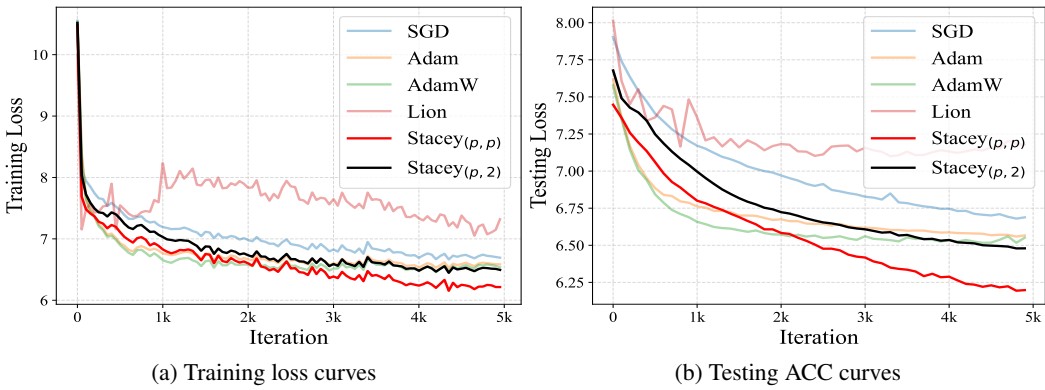

(a) Training loss curves

(b) Testing ACC curves

Figure 4: Learning curves of LLM pretraining at the first 5000 iterations among different optimizers.

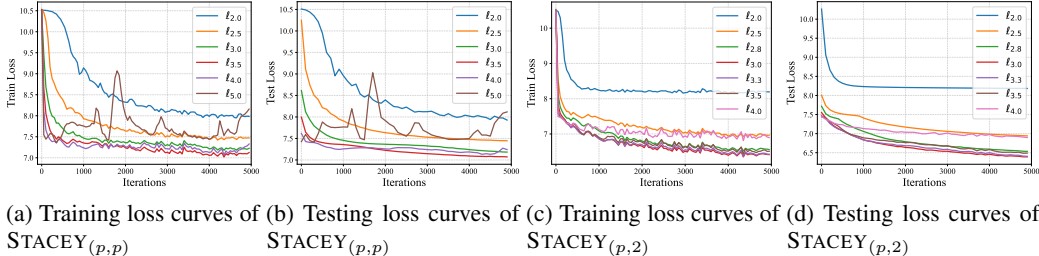

(a) Training loss curves of STACEY$_{(p,p)}$ (b) Testing loss curves of STACEY$_{(p,p)}$ (c) Training loss curves of STACEY$_{(p,2)}$ (d) Testing loss curves of STACEY$_{(p,2)}$

Figure 5: Learning curves of LLM pretraining at the first 5000 iterations with varying $\ell_p$-norm.

## 5.4 DISCUSSION

As we observe throughout the experiments, STACEY demonstrates superior performance over SGD, which showcases its ability to adapt to a broader range of non-Euclidean geometries. This adaptability verifies STACEY's convergence for general $\ell_p$-norms, making it a better choice for optimization tasks that present complex geometries and extend beyond the conventional Euclidean frameworks. Compared with Adam and AdamW, STACEY confirms that the introduced acceleration technique is well-aligned with the principles of non-Euclidean optimization. The superior results validate that STACEY's acceleration mechanism, which is purposefully designed for non-Euclidean spaces, outperforms the traditional adaptive methods that rely on Euclidean-centric assumptions. Furthermore, STACEY's improved performance over Lion highlights the effectiveness of interpolating primal and dual sequences as an acceleration strategy, in contrast to simply incorporating momentum. The primal-dual interpolation ensures a more balanced and stable progression towards optimality, leveraging information from both primal and dual sequences. This strategy allows STACEY to achieve faster convergence, even in challenging settings and complex tasks like large-scale image classification and pretraining LLMs.

## 6 CONCLUSION

This paper investigates the steepest descent algorithm in $\ell_p$ norm for stochastic non-convex optimization. We establish for the stochastic $\ell_p$ descent algorithm an $O(\epsilon^{-4})$ convergence rate in expectation to a stationary point with respect to the dual norm $\|\cdot\|_{p^*}^{p^*}$. Building on these techniques, we further proposed an acceleration scheme for non-Euclidean methods, incorporated stochastic $\ell_p$ descent with mirror descent, and presented an accelerated algorithm called STACEY. We evaluated the performance of STACEY on large-scale image classification and pretraining language modeling tasks and achieved both faster convergence and higher accuracy compared to other methods.

## REPRODUCIBILITY STATEMENT

The reproducibility of our research is ensured through two key measures. Firstly, the algorithm proposed in this paper has been explicitly described in detail in the appendix, allowing for a clear understanding of our approach. Secondly, to facilitate direct replication of our work, we have provided the complete implementations as anonymously downloadable source code in the supplementary materials. These measures should enable other researchers to fully reproduce and validate our findings.

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

# A PROOFS

## A.1 COMPLETE PROOF FOR THEOREM 1

**Theorem 1** (Main). *Running Algorithm 1 on some (possibly non-convex) function $f$ that satisfies Assumptions 1 to 4 yields*

$$\mathbb{E}\left[\frac{1}{T}\sum_{t=0}^{T-1}\|\nabla f(\theta_t)\|_{p^*}^{p^*}\right] \leq \frac{f(\theta_0) - f(\theta^*)}{\eta T} + \frac{1}{T}\sum_{t=0}^{T-1}\frac{\frac{2p-1}{p-1}G^{\frac{1}{p-1}}\|\vec{\sigma}\|_1}{\sqrt{n_t}} + \frac{L\eta G^{\frac{2}{p-1}}}{2}$$

*where $n_t$ is the batch size in iteration $t$ and $L$, $\vec{\sigma}$, and $G$ are constants from Assumption 1, 3, 4. Further letting the batch size $n_t = T$, the number of gradient call is $N = T^2$ for $T$ iterations. With $\eta = \frac{1}{L^{\frac{1}{2}}G^{\frac{1}{p-1}}T^{\frac{1}{2}}}$ we have*

$$\mathbb{E}\left[\frac{1}{T}\sum_{t=0}^{T-1}\|\nabla f(\theta_t)\|_{p^*}^{p^*}\right] \leq \frac{1}{N^{\frac{1}{4}}}\left[L^{\frac{1}{2}}G^{\frac{1}{p-1}}\left(f(\theta_0) - f(\theta^*) + \frac{1}{2}\right) + \frac{2p-1}{p-1}G^{\frac{1}{p-1}}\|\vec{\sigma}\|_1\right],$$

*i.e., Algorithm 1 takes $N \in \mathcal{O}\left(\epsilon^{-4}\right)$ gradient queries to reach an $\epsilon$-approximate stationary point.*

*Proof.* Starting with Assumption 1 and the descent step in Algorithm 1,

$$f(\theta_{t+1}) \leq f(\theta_t) + \langle\nabla f(\theta_t), \theta_{t+1} - \theta_t\rangle + \frac{L}{2}\|\theta_{t+1} - \theta_t\|_p^2$$

$$= f(\theta_t) + \eta\langle\nabla f(\theta_t), -s(g(\theta_t))\rangle + \frac{L}{2}\|s(g(\theta_t))\|_p^2$$

$$= f(\theta_t) - \underbrace{\eta\langle\nabla f(\theta_t), s(\nabla f(\theta_t))\rangle}_{A}$$

$$+ \underbrace{\eta\langle\nabla f(\theta_t), s(\nabla f(\theta_t)) - s(g(\theta_t))\rangle}_{B} + \underbrace{\frac{L\eta^2}{2}\|s(g(\theta_t))\|_p^2}_{C}$$

Now we analyze these terms one by one.

$$A = \sum_{i=1}^{d}\nabla f(\theta_t)^{(i)} \cdot \frac{\nabla f(\theta_t)^{(i)}}{|\nabla f(\theta_t)^{(i)}|^{\frac{p-2}{p-1}}}$$

$$= \sum_{i=1}^{d}|\nabla f(\theta_t)^{(i)}|^{\frac{p}{p-1}}$$

$$= \|\nabla f(\theta_t)\|_{p^*}^{p^*}$$

For term $B$,

$$B = \eta\sum_{i=1}^{d}\nabla f(\theta_t)^{(i)}\left(\frac{\nabla f(\theta_t)^{(i)}}{|\nabla f(\theta_t)^{(i)}|^{\frac{p-2}{p-1}}} - \frac{g(\theta_t)^{(i)}}{|g(\theta_t)^{(i)}|^{\frac{p-2}{p-1}}}\right)$$

$$= \eta\sum_{i=1}^{d}\nabla f(\theta_t)^{(i)}\left(\text{sign}\left(\nabla f(\theta_t)^{(i)}\right)|\nabla f(\theta_t)^{(i)}|^{\frac{1}{p-1}} - \text{sign}\left(g(\theta_t)^{(i)}\right)|g(\theta_t)^{(i)}|^{\frac{1}{p-1}}\right)$$

$$\leq \eta\sum_{i=1}^{d}\left|\nabla f(\theta_t)^{(i)}\right|\left|\text{sign}\left(\nabla f(\theta_t)^{(i)}\right)|\nabla f(\theta_t)^{(i)}|^{\frac{1}{p-1}} - \text{sign}\left(g(\theta_t)^{(i)}\right)|g(\theta_t)^{(i)}|^{\frac{1}{p-1}}\right|$$

$$= \underbrace{\eta\sum_{i=1}^{d}\left|\nabla f(\theta_t)^{(i)}\right|\left(|\nabla f(\theta_t)^{(i)}|^{\frac{1}{p-1}} + |g(\theta_t)^{(i)}|^{\frac{1}{p-1}}\right)\mathbb{I}\left[\text{sign}\left(\nabla f(\theta_t)^{(i)}\right) \neq \text{sign}\left(g(\theta_t)^{(i)}\right)\right]}_{B_1}$$

$$+ \underbrace{\eta\sum_{i=1}^{d}\left|\nabla f(\theta_t)^{(i)}\right|\left||\nabla f(\theta_t)^{(i)}|^{\frac{1}{p-1}} - |g(\theta_t)^{(i)}|^{\frac{1}{p-1}}\right|\mathbb{I}\left[\text{sign}\left(\nabla f(\theta_t)^{(i)}\right) = \text{sign}\left(g(\theta_t)^{(i)}\right)\right]}_{B_2}$$

$B_1$ is bounded in expectation by $\frac{2\eta G^{\frac{1}{p-1}}\|\vec{\sigma}\|_1}{\sqrt{n_t}}$ in Lemma 2 and $B_2$ is bounded in expectation by $\frac{\eta G^{\frac{1}{p-1}}\|\vec{\sigma}\|_1}{(p-1)\sqrt{n_t}}$ in Lemma 3.

$$
\begin{aligned}
C &= \frac{L\eta^2}{2}\left(\sum_{i=1}^d \left|\frac{\nabla f(\theta_t)^{(i)}}{|\nabla f(\theta_t)^{(i)}|^{\frac{p-2}{p-1}}}\right|^p\right)^{\frac{2}{p}} \\
&= \frac{L\eta^2}{2}\left(\sum_{i=1}^d \left|\nabla f(\theta_t)^{(i)}\right|^{\frac{p}{p-1}}\right)^{\frac{2}{p}} \\
&= \frac{L\eta^2}{2}\|\nabla f(\theta_t)\|_{p^*}^{\frac{2}{p-1}} \\
&\leq \frac{L\eta^2 G^{\frac{2}{p-1}}}{2}
\end{aligned}
$$

Therefore,

$$
\eta\mathbb{E}\left[\|\nabla f(\theta_t)\|_{p^*}^{p^*}\right] \leq f(\theta_t) - f(\theta_{t+1}) + \frac{\eta(2p-1)G^{\frac{1}{p-1}}\|\vec{\sigma}\|_1}{(p-1)\sqrt{n_t}} + \frac{L\eta^2 G^{\frac{2}{p-1}}}{2}
$$

By telescoping through $t = 0, \cdots, T-1$, we get

$$
\mathbb{E}\left[\frac{1}{T}\sum_{t=0}^{T-1}\|\nabla f(\theta_t)\|_{p^*}^{p^*}\right] \leq \frac{f(\theta_0)-f(\theta_T)}{\eta T} + \frac{1}{T}\sum_{t=0}^{T-1}\frac{(2p-1)G^{\frac{1}{p-1}}\|\vec{\sigma}\|_1}{(p-1)\sqrt{n_t}} + \frac{L\eta G^{\frac{2}{p-1}}}{2}
$$

$\square$

**Lemma 2.**

$$
\mathbb{E}\left[\eta\sum_{i=1}^d \left|\nabla f(\theta_t)^{(i)}\right|\left(|\nabla f(\theta_t)^{(i)}|^{\frac{1}{p-1}} + |g(\theta_t)^{(i)}|^{\frac{1}{p-1}}\right)\mathbb{I}\left[\operatorname{sign}\left(\nabla f(\theta_t)^{(i)}\right) \neq \operatorname{sign}\left(g(\theta_t)^{(i)}\right)\right]\right] \leq \frac{2\eta G^{\frac{1}{p-1}}\|\vec{\sigma}\|_1}{\sqrt{n_t}}
$$

*Proof.* By Corollary 3 (b),

$$
\mathbb{E}\left[\eta\sum_{i=1}^d \left|\nabla f(\theta_t)^{(i)}\right|\left(|\nabla f(\theta_t)^{(i)}|^{\frac{1}{p-1}} + |g(\theta_t)^{(i)}|^{\frac{1}{p-1}}\right)\mathbb{I}\left[\operatorname{sign}\left(\nabla f(\theta_t)^{(i)}\right) \neq \operatorname{sign}\left(g(\theta_t)^{(i)}\right)\right]\right]
$$

$$
\leq 2\eta G^{\frac{1}{p-1}}\mathbb{E}\left[\sum_{i=1}^d \left|\nabla f(\theta_t)^{(i)}\right|\mathbb{I}\left[\operatorname{sign}\left(\nabla f(\theta_t)^{(i)}\right) \neq \operatorname{sign}\left(g(\theta_t)^{(i)}\right)\right]\right]
$$

$$
= 2\eta G^{\frac{1}{p-1}}\sum_{i=1}^d \left|\nabla f(\theta_t)^{(i)}\right|\mathbb{P}\left[\operatorname{sign}\left(\nabla f(\theta_t)^{(i)}\right) \neq \operatorname{sign}\left(g(\theta_t)^{(i)}\right)\right]
$$

$$
\leq 2\eta G^{\frac{1}{p-1}}\sum_{i=1}^d \left|\nabla f(\theta_t)^{(i)}\right|\mathbb{P}\left[\left|g(\theta_t)^{(i)} - \nabla f(\theta_t)^{(i)}\right| \geq \left|\nabla f(\theta_t)^{(i)}\right|\right]
$$

$$
\leq 2\eta G^{\frac{1}{p-1}}\sum_{i=1}^d \left|\nabla f(\theta_t)^{(i)}\right|\frac{\mathbb{E}\left[\left|g(\theta_t)^{(i)} - \nabla f(\theta_t)^{(i)}\right|\right]}{|\nabla f(\theta_t)^{(i)}|}
$$

$$
\leq 2\eta G^{\frac{1}{p-1}}\sum_{i=1}^d \sqrt{\mathbb{E}\left[|g(\theta_t)^{(i)} - \nabla f(\theta_t)^{(i)}|^2\right]}
$$

$$
\leq \frac{2\eta G^{\frac{1}{p-1}}\sum_{i=1}^d \sigma_i}{\sqrt{n_t}}
$$

$$
= \frac{2\eta G^{\frac{1}{p-1}}\|\vec{\sigma}\|_1}{\sqrt{n_t}}
$$

where for the last three inequalities we used Markov's inequality, Jensen's inequality, and Assumption 3. $\square$

**Lemma 3.**

$$
\mathbb{E}\left[\eta\sum_{i=1}^d \left|\nabla f(\theta_t)^{(i)}\right|\left||\nabla f(\theta_t)^{(i)}|^{\frac{1}{p-1}} - |g(\theta_t)^{(i)}|^{\frac{1}{p-1}}\right|\mathbb{I}\left[\operatorname{sign}\left(\nabla f(\theta_t)^{(i)}\right) = \operatorname{sign}\left(g(\theta_t)^{(i)}\right)\right]\right] \leq \frac{\eta G^{\frac{1}{p-1}}\|\vec{\sigma}\|_1}{(p-1)\sqrt{n_t}}
$$

*Proof.* Denoting $\mathbb{E}\left[\cdot \mid \text{sign}\left(\nabla f(\theta_t)^{(i)}\right) = \text{sign}\left(g(\theta_t)^{(i)}\right)\right]$ as $\mathbb{E}_{|=}[\cdot]$, and $\mathbb{P}\left[\text{sign}\left(\nabla f(\theta_t)^{(i)}\right) = \text{sign}\left(g(\theta_t)^{(i)}\right)\right]$ as $\mathbb{P}[=]$,

$$\mathbb{E}\left[\eta \sum_{i=1}^{d} \left|\nabla f(\theta_t)^{(i)}\right| \left| |\nabla f(\theta_t)^{(i)}|^{\frac{1}{p-1}} - |g(\theta_t)^{(i)}|^{\frac{1}{p-1}} \right| \mathbb{I}\left[\text{sign}\left(\nabla f(\theta_t)^{(i)}\right) = \text{sign}\left(g(\theta_t)^{(i)}\right)\right]\right]$$

$$= \eta \mathbb{E}_{|=}\left[\sum_{i=1}^{d} \left|\nabla f(\theta_t)^{(i)}\right| \left| |\nabla f(\theta_t)^{(i)}|^{\frac{1}{p-1}} - |g(\theta_t)^{(i)}|^{\frac{1}{p-1}} \right|\right] \mathbb{P}[=]$$

$$= \eta \mathbb{E}_{|=}\left[\sum_{i=1}^{d} \left|\nabla f(\theta_t)^{(i)}\right| \left| |\nabla f(\theta_t)^{(i)}|^{\frac{1}{p-1}} - |g(\theta_t)^{(i)}|^{\frac{1}{p-1}} \right|\right] \mathbb{P}[=]$$

$$= \eta \mathbb{E}_{|=}\left[\sum_{i=1}^{d} \left|\nabla f(\theta_t)^{(i)}\right| \left| \left(|g(\theta_t)^{(i)}|^{\frac{1}{p-1}} + \frac{1}{p-1}\text{sign}(\zeta^{(i)})\left|\zeta^{(i)}\right|^{\frac{2-p}{p-1}}\left(\nabla f(\theta_t)^{(i)} - g(\theta_t)^{(i)}\right)\right) - |g(\theta_t)^{(i)}|^{\frac{1}{p-1}} \right|\right] \mathbb{P}[=]$$

$$= \eta \mathbb{E}_{|=}\left[\sum_{i=1}^{d} \left|\nabla f(\theta_t)^{(i)}\right| \left| \frac{1}{p-1}\text{sign}(\zeta^{(i)})\left|\zeta^{(i)}\right|^{\frac{2-p}{p-1}}\left(\nabla f(\theta_t)^{(i)} - g(\theta_t)^{(i)}\right) \right|\right] \mathbb{P}[=]$$

$$= \frac{\eta}{p-1} \mathbb{E}_{|=}\left[\sum_{i=1}^{d} \left|\nabla f(\theta_t)^{(i)}\right| \left|\zeta^{(i)}\right|^{\frac{2-p}{p-1}} \left|\nabla f(\theta_t)^{(i)} - g(\theta_t)^{(i)}\right|\right] \mathbb{P}[=],$$

in which the second equality holds by taking the zeroth order Taylor expansion of $\left|\nabla f(\theta_t)^{(i)}\right|^{\frac{1}{p-1}}$ at $g(\theta_t)^{(i)}$ with Lagrange remainder, and $\zeta^{(i)}$ is in the range from $\nabla f(\theta_t)^{(i)}$ to $g(\theta_t)^{(i)}$.

Given $\text{sign}\left(\nabla f(\theta_t)^{(i)}\right) = \text{sign}\left(g(\theta_t)^{(i)}\right)$, by the definition of $\zeta^{(i)}$ in the Lagrange remainder, we must have either $\left|\nabla f(\theta_t)^{(i)}\right| \leq \left|\zeta^{(i)}\right| \leq \left|g(\theta_t)^{(i)}\right|$ or $\left|\nabla f(\theta_t)^{(i)}\right| \geq \left|\zeta^{(i)}\right| \geq \left|g(\theta_t)^{(i)}\right|$. Now we analyze these two cases respectively. We write out the derivations separately for clarity and simplicity, alternatively one can merge these two cases with the law of total expectation.

(1) If $\left|\nabla f(\theta_t)^{(i)}\right| \leq \left|\zeta^{(i)}\right| \leq \left|g(\theta_t)^{(i)}\right|$, then

$$\frac{\eta}{p-1} \mathbb{E}_{|=}\left[\sum_{i=1}^{d} \left|\nabla f(\theta_t)^{(i)}\right| \left|\zeta^{(i)}\right|^{\frac{2-p}{p-1}} \left|\nabla f(\theta_t)^{(i)} - g(\theta_t)^{(i)}\right|\right] \mathbb{P}[=]$$

$$\leq \frac{\eta}{p-1} \mathbb{E}_{|=}\left[\sum_{i=1}^{d} \left|\zeta^{(i)}\right| \left|\zeta^{(i)}\right|^{\frac{2-p}{p-1}} \left|\nabla f(\theta_t)^{(i)} - g(\theta_t)^{(i)}\right|\right] \mathbb{P}[=]$$

$$= \frac{\eta}{p-1} \mathbb{E}_{|=}\left[\sum_{i=1}^{d} \left|\zeta^{(i)}\right|^{\frac{1}{p-1}} \left|\nabla f(\theta_t)^{(i)} - g(\theta_t)^{(i)}\right|\right] \mathbb{P}[=]$$

$$\leq \frac{\eta}{p-1} \mathbb{E}_{|=}\left[\sum_{i=1}^{d} \left|g(\theta_t)^{(i)}\right|^{\frac{1}{p-1}} \left|\nabla f(\theta_t)^{(i)} - g(\theta_t)^{(i)}\right|\right] \mathbb{P}[=]$$

$$\leq \frac{\eta G^{\frac{1}{p-1}}}{p-1} \sum_{i=1}^{d} \mathbb{E}_{|=}\left[\left|\nabla f(\theta_t)^{(i)} - g(\theta_t)^{(i)}\right|\right] \mathbb{P}[=]$$

$$= \frac{\eta G^{\frac{1}{p-1}}}{p-1} \sum_{i=1}^{d} \frac{\mathbb{E}\left[\left|\nabla f(\theta_t)^{(i)} - g(\theta_t)^{(i)}\right|\right]}{\mathbb{P}[=]} \mathbb{P}[=]$$

$$\leq \frac{\eta G^{\frac{1}{p-1}}}{p-1} \sum_{i=1}^{d} \sqrt{\mathbb{E}\left[|\nabla f(\theta_t)^{(i)} - g(\theta_t)^{(i)}|^2\right]} \qquad \text{(Jensen's)}$$

$$\leq \frac{\eta G^{\frac{1}{p-1}}}{p-1} \sum_{i=1}^{d} \frac{\sigma_i}{\sqrt{n_t}} \qquad \text{(Assumption 3)}$$

$$= \frac{\eta G^{\frac{1}{p-1}} \|\vec{\sigma}\|_1}{(p-1)\sqrt{n_t}}$$

(2) If $\left|\nabla f(\theta_t)^{(i)}\right| \geq \left|\zeta^{(i)}\right| \geq \left|g(\theta_t)^{(i)}\right|$, then

$$\frac{\eta}{p-1}\mathbb{E}_{|=}\left[\sum_{i=1}^{d}\left|\nabla f(\theta_t)^{(i)}\right|\left|\zeta^{(i)}\right|^{\frac{2-p}{p-1}}\left|\nabla f(\theta_t)^{(i)}-g(\theta_t)^{(i)}\right|\right]\mathbb{P}\left[=\right]$$

$$\leq \frac{\eta}{p-1}\mathbb{E}_{|=}\left[\sum_{i=1}^{d}\left|\nabla f(\theta_t)^{(i)}\right|\left|g(\theta_t)^{(i)}\right|^{\frac{2-p}{p-1}}\left|\nabla f(\theta_t)^{(i)}-g(\theta_t)^{(i)}\right|\right]\mathbb{P}\left[=\right]$$

$$\leq \frac{\eta}{(p-1)\mathbb{P}[=]}\mathbb{E}\left[\sum_{i=1}^{d}\left|\nabla f(\theta_t)^{(i)}\right|\left|g(\theta_t)^{(i)}\right|^{\frac{2-p}{p-1}}\left|\nabla f(\theta_t)^{(i)}-g(\theta_t)^{(i)}\right|\right]\mathbb{P}\left[=\right]$$

$$\leq \frac{\eta}{p-1}\sum_{i=1}^{d}\sqrt{\mathbb{E}\left[|\nabla f(\theta_t)^{(i)}|^2|g(\theta_t)^{(i)}|^{\frac{2(2-p)}{p-1}}\right]\mathbb{E}\left[|\nabla f(\theta_t)^{(i)}-g(\theta_t)^{(i)}|^2\right]} \qquad \text{(Cauchy-Schwarz)}$$

$$\leq \frac{\eta}{p-1}\sum_{i=1}^{d}\sqrt{|\nabla f(\theta_t)^{(i)}|^2\,\mathbb{E}\left[|g(\theta_t)^{(i)}|^{\frac{2(2-p)}{p-1}}\right]\frac{\sigma_i^2}{n_t}} \qquad \text{(Assumption 3)}$$

$$\leq \frac{\eta}{p-1}\sum_{i=1}^{d}\sqrt{|\nabla f(\theta_t)^{(i)}|^2\left(\mathbb{E}\left[|g(\theta_t)^{(i)}|^2\right]\right)^{\frac{2-p}{p-1}}\frac{\sigma_i^2}{n_t}} \qquad \text{(Jensen's)}$$

$$\leq \frac{\eta}{p-1}\sum_{i=1}^{d}\sqrt{|\nabla f(\theta_t)^{(i)}|^2\left(\mathrm{Var}\left[g(\theta_t)^{(i)}\right]+(\mathbb{E}\left[g(\theta_t)^{(i)}\right])^2\right)^{\frac{2-p}{p-1}}\frac{\sigma_i^2}{n_t}} \qquad \text{(Variance Definition)}$$

$$\leq \frac{\eta}{p-1}\sum_{i=1}^{d}\sqrt{|\nabla f(\theta_t)^{(i)}|^2\left(\mathbb{E}\left[g(\theta_t)^{(i)}\right]\right)^{\frac{2(2-p)}{p-1}}\frac{\sigma_i^2}{n_t}}$$

$$= \frac{\eta}{p-1}\sum_{i=1}^{d}\left|\nabla f(\theta_t)^{(i)}\right|^{\frac{1}{p-1}}\frac{\sigma_i}{\sqrt{n_t}} \qquad \text{(Assumption 2)}$$

$$\leq \frac{\eta G^{\frac{1}{p-1}}\|\vec{\sigma}\|_1}{(p-1)\sqrt{n_t}}.$$

Combining these two cases together (e.g. by the law of total expectation) completes the proof.

□

## B $\ell_2$ MAJORIZATION AND $\ell_p$ SMOOTHNESS

Another assumption of interest, as studied by Bernstein et al. (2018) (as well as Karimi et al. (2016)), is that of $\ell_2$ *majorization* (with respect to $\vec{L} = [L_1, \ldots, L_d]$), meaning that for all $x, y \in \mathbb{R}^d$,

$$\left|f(y) - f(x) - \nabla f(x)^\top(y-x)\right| \leq \frac{1}{2}\sum_{i=1}^{d}L_i(y^{(i)}-x^{(i)})^2.$$

We may equivalently express this condition as 1-smoothness w.r.t. $\|\cdot\|_{\mathbf{L}}$, where $\mathbf{L} := \mathrm{diag}(\vec{L})$, i.e., for all $x, y \in \mathbb{R}^d$, $\|\nabla f(y) - \nabla f(x)\|_{\mathbf{L}^{-1}} \leq \|y-x\|_{\mathbf{L}}$.

Interestingly, we may observe that, for any $1 \leq p \leq \infty$,

$$\frac{1}{\|\vec{L}\|_{p^*}^{1/2}}\|\nabla f(y) - \nabla f(x)\|_{2p/(2p-1)} \leq \|\nabla f(y) - \nabla f(x)\|_{\mathbf{L}^{-1}} \leq \|y-x\|_{\mathbf{L}} \leq \|\vec{L}\|_{p^*}^{1/2}\|y-x\|_{2p},$$

where the first inequality holds by reverse Hölder's inequality, i.e., for $u, v \in \mathbb{R}^d$, $\sum_{i=1}^{d}|u^{(i)}v^{(i)}| \geq \|u\|_{1/q}\|v\|_{\frac{-1}{q-1}}$ (where we choose $q = \frac{2p-1}{p}$), and the last inequality holds by Hölder's inequality.

Rearranging, we have $\|\nabla f(y) - \nabla f(x)\|_{2p/(2p-1)} \leq \|\vec{L}\|_{p^*}\|y-x\|_{2p}$, and so it follows that $\ell_2$ majorization implies $\|\vec{L}\|_{\frac{p}{p-2}}$-smoothness w.r.t. $\|\cdot\|_p$. Thus, while this condition is sufficient to entail $\ell_p$ smoothness (as previously noted by Balles et al. (2020) in the case of $p = \infty$), we nevertheless prefer to work directly with $\ell_p$ smoothness assumptions, as we believe they provide a more natural pairing for the methods we consider.

Table 3: Image classification on CIFAR at the 50th, 100th, and 200th epochs. $\text{STACEY}_{(2,p)}$ consistently lower performance than $\text{STACEY}_{(p,p)}$ at all epochs.

| Optimizer | Training NLL ↓ | | | Testing ACC (%) ↑ | | |
|---|---|---|---|---|---|---|
| | @50 epoch | @100 epoch | @200 epoch | @50 epoch | @100 epoch | @200 epoch |
| $\text{STACEY}_{(2,p)}$ | 0.1017 | 0.0365 | 0.0083 | 90.78 | 91.88 | 93.55 |
| $\text{STACEY}_{(p,p)}$ | 0.1438 | 0.0405 | 0.0006 | 88.95 | 91.50 | **94.05** |
| $\text{STACEY}_{(p,2)}$ | **0.0375** | **0.0104** | **0.0005** | **91.87** | **92.92** | 93.99 |

Table 4: Image classification on ImageNet at the 20th, 50th, and 90th epochs. $\text{STACEY}_{(2,p)}$ consistently lower performance than $\text{STACEY}_{(p,p)}$ at all epochs.

| Optimizer | Training NLL ↓ | | | Testing Top-1 ACC (%) ↑ | | |
|---|---|---|---|---|---|---|
| | @20 epoch | @50 epoch | @90 epoch | @20 epoch | @50 epoch | @90 epoch |
| $\text{STACEY}_{(2,p)}$ | 2.5178 | 1.8038 | 1.4274 | 50.59 | 61.72 | 65.11 |
| $\text{STACEY}_{(p,p)}$ | **1.9371** | **1.2064** | **0.9902** | **60.84** | **68.23** | **69.88** |
| $\text{STACEY}_{(p,2)}$ | 3.3706 | 2.5149 | 2.1975 | 32.16 | 49.39 | 57.33 |

## C  THE VARIATION $\text{STACEY}_{(2,p)}$

For the sake of completion, we also considered the (natural) variant $\text{STACEY}_{(2,p)}$, which couples $\ell_2$ steepest descent with mirror descent (for dgf $\frac{1}{p}\|\cdot\|_p^p$).

---

**Algorithm 4** $\text{STACEY}_{(2,p)}$ Optimizer

**input** $p, \beta_1, \beta_2, \alpha, \tau, \eta, \epsilon, \lambda, f$
**initialize** $\theta_0, z_0, m_0 \leftarrow 0$
1: **while** $\theta_{t+1}$ not converged **do**
2:   $g_t \leftarrow g(\theta_t)$
3:   $c_{t+1} \leftarrow \beta_1 m_t + (1 - \beta_1)g_t$
4:   $y_{t+1} \leftarrow \theta_t - \eta_t c_{t+1}$
5:   $z_{t+1}^{(i)} = \dfrac{\left|z_t^{(i)}\right|^{p-2}z_t^{(i)} - \alpha c_{t+1}^{(i)}}{\left|\left|z_t^{(i)}\right|^{p-2}z_t^{(i)} - \alpha c_{t+1}^{(i)}\right|^{\frac{p-2}{p-1}}}, \forall i \in [d]$
6:   $\theta_{t+1} = \tau z_{t+1} + (1 - \tau)y_{t+1} - \eta_t \lambda \theta_t$
7:   $m_{t+1} = \beta_2 m_t + (1 - \beta_2)g_t$
**return** $\theta_{t+1}$

---

Table 3&4 show the classification results of $\text{STACEY}_{(2,p)}$ optimizer. The experimental results of $\text{STACEY}_{(2,p)}$ optimizer with varying $p$-norm are shown in Appendix E. Specifically, the results on CIFAR (Krizhevsky, 2009) are shown in Fig. 6a&7a, the results on ImageNet (Deng et al., 2009) are shown in Fig. 8a&9a&10a, and the results on LLM pertaining are shown in Fig. 11a&12a&13a.

## D  HYPER-PARAMETER CHOICES

We list the hyper-parameters used in the experiments in Table 5&6&7, which are determined by grid search. We employ Weight & Bias platform[10] to tune the hyper-parameters. To ensure a fair comparison, the experimental settings beyond the listed hyper-parameters remain the same for all optimizers. For example, the data augmentation for ImageNet (Deng et al., 2009) and CIFAR (Krizhevsky, 2009) is random cropping plus random horizontal flipping.

---

[10]https://github.com/wandb/wandb.

Table 5: CIFAR hyper-parameters.

| Model | Optimizer | Batch Size | $p$ | $lr$ $(\eta)$ | $\alpha$ | $\beta_1$ | $\beta_2$ | $\lambda$ | $\tau$ | $\epsilon$ |
|---|---|---|---|---|---|---|---|---|---|---|
| ResNet-18 | SGD w/ Nesterov | 128 | - | 0.02 | - | 0.9 | - | 0.0002 | - | - |
| ResNet-18 | Adam | 128 | - | 0.001 | - | 0.9 | 0.999 | 0.0005 | - | 1e-8 |
| ResNet-18 | AdamW | 128 | - | 0.01 | - | 0.9 | 0.999 | 0.0005 | - | 1e-8 |
| ResNet-18 | Lion | 128 | - | 0.001 | - | 0.9 | 0.99 | 0.01 | - | - |
| ResNet-18 | STACEY$_{(2,p)}$ | 128 | 3.5 | 0.02 | 0.01 | 0.9 | 0.99 | 0.4 | 0.001 | - |
| ResNet-18 | STACEY$_{(p,p)}$ | 128 | 2 | 0.1 | 0.1 | 0.9 | 0.99 | 0.01 | 0.001 | 1e-12 |
| ResNet-18 | STACEY$_{(p,2)}$ | 128 | 2 | 0.1 | 0.1 | 0.9 | 0.99 | 0.01 | 0.001 | 1e-12 |

Table 6: ImageNet hyper-parameters.

| Model | Optimizer | Batch Size | $p$ | $lr$ $(\eta)$ | $\alpha$ | $\beta_1$ | $\beta_2$ | $\lambda$ | $\tau$ | $\epsilon$ |
|---|---|---|---|---|---|---|---|---|---|---|
| ResNet-50 | SGD w/ Nesterov | 256 | - | 0.01 | - | - | - | 0.0005 | - | - |
| ResNet-50 | STACEY$_{(2,p)}$ | 256 | 2.2 | 0.01 | 0.01 | 0.9 | 0.99 | 0.0005 | 0.001 | - |
| ResNet-50 | STACEY$_{(p,p)}$ | 256 | 3 | 0.01 | 0.1 | 0.9 | 0.99 | 0.0005 | 0.001 | 1e-8 |
| ResNet-50 | STACEY$_{(p,2)}$ | 256 | 2.8 | 0.01 | 0.01 | 0.9 | 0.99 | 0.0005 | 0.001 | 1e-8 |

# E  ADDITIONAL EXPERIMENTAL RESULTS

## E.1  LEARNING CURVES OF VARYING $\ell_p$-NORM ON CIFAR CLASSIFICATION

The results are shown in Fig. 6&7.

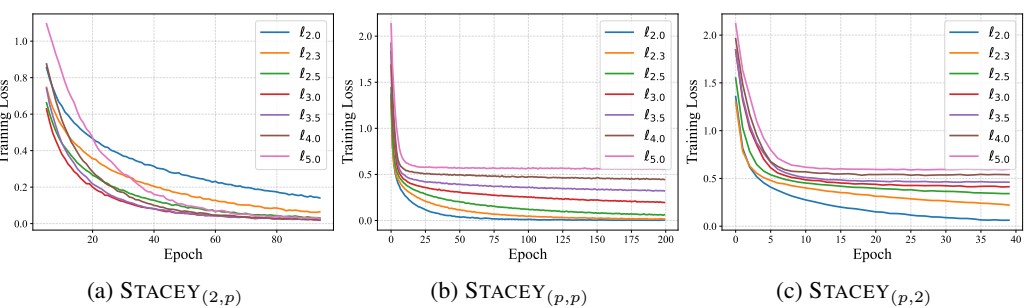

(a) STACEY$_{(2,p)}$      (b) STACEY$_{(p,p)}$      (c) STACEY$_{(p,2)}$

Figure 6: Training loss of CIFAR classification with varying $\ell_p$-norm.

## E.2  LEARNING CURVES OF VARYING $\ell_p$-NORM ON IMAGENET CLASSIFICATION

The results are shown in Fig. 8&9&10.

## E.3  LEARNING CURVES OF VARYING $\ell_p$-NORM ON LLM PRETRAINING

The results are shown in Fig. 11&12&13.

Table 7: Hyper-parameters for LLM pretraining.

| Model | Optimizer | Batch Size | $p$ | $lr$ ($\eta$) | $\alpha$ | $\beta_1$ | $\beta_2$ | $\lambda$ | $\tau$ | $\epsilon$ |
|---|---|---|---|---|---|---|---|---|---|---|
| Llama 100M | SGD | 16 | - | 0.01 | - | - | - | 0.0005 | - | - |
| Llama 100M | Adam | 16 | - | 0.0001 | - | 0.9 | 0.999 | 0.01 | - | 1e-8 |
| Llama 100M | AdamW | 16 | - | 0.0001 | - | 0.9 | 0.999 | 0.05 | - | 1e-8 |
| Llama 100M | Lion | 16 | - | 0.05 | - | 0.9 | 0.999 | 0.01 | - | - |
| Llama 100M | STACEY$_{(2,p)}$ | 16 | 2.8 | 0.05 | 0.01 | 0.9 | 0.99 | 0.01 | 0.001 | - |
| Llama 100M | STACEY$_{(p,p)}$ | 16 | 3 | 0.01 | 0.01 | 0.9 | 0.99 | 0.01 | 0.001 | 1e-8 |
| Llama 100M | STACEY$_{(p,2)}$ | 16 | 2.8 | 0.01 | 0.01 | 0.9 | 0.99 | 0.0005 | 0.001 | 1e-8 |

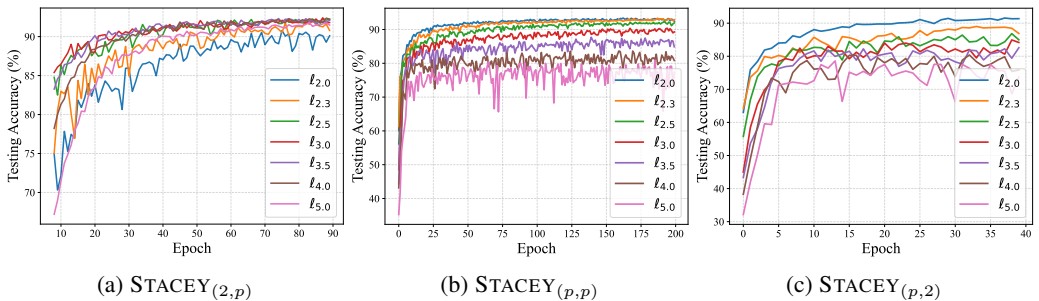

(a) STACEY$_{(2,p)}$     (b) STACEY$_{(p,p)}$     (c) STACEY$_{(p,2)}$

Figure 7: Testing accuracy of CIFAR classification with varying $\ell_p$-norm.

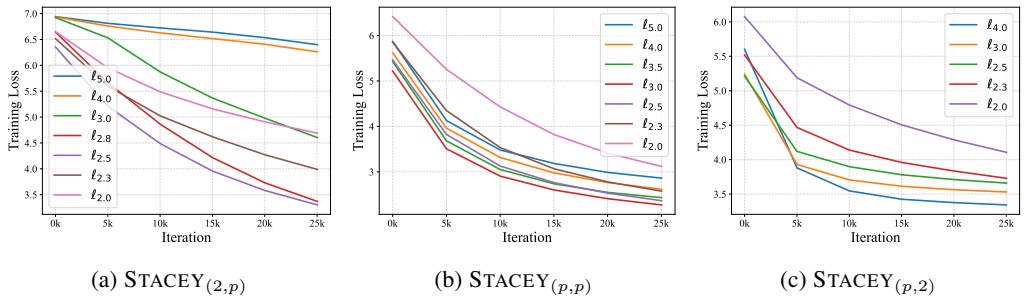

(a) STACEY$_{(2,p)}$     (b) STACEY$_{(p,p)}$     (c) STACEY$_{(p,2)}$

Figure 8: Training loss of ImageNet classification with varying $\ell_p$-norm.

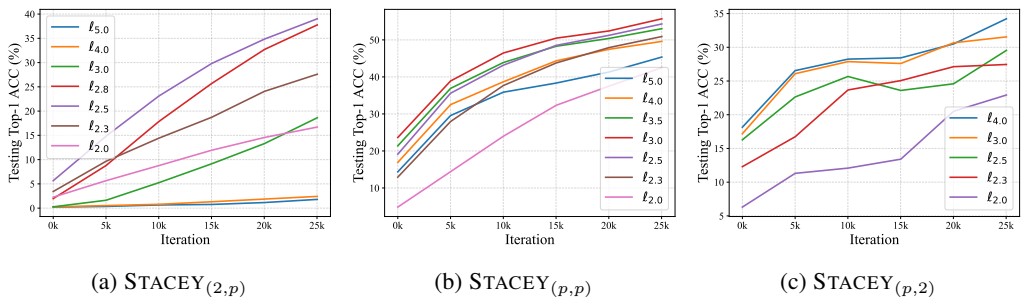

(a) STACEY$_{(2,p)}$     (b) STACEY$_{(p,p)}$     (c) STACEY$_{(p,2)}$

Figure 9: Testing Top-1 accuracy of ImageNet classification with varying $\ell_p$-norm.

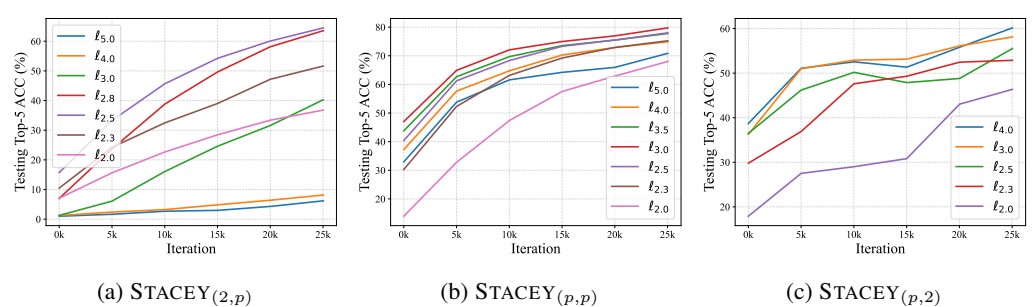

(a) STACEY$_{(2,p)}$      (b) STACEY$_{(p,p)}$      (c) STACEY$_{(p,2)}$

Figure 10: Testing Top-5 accuracy of ImageNet classification with varying $\ell_p$-norm.

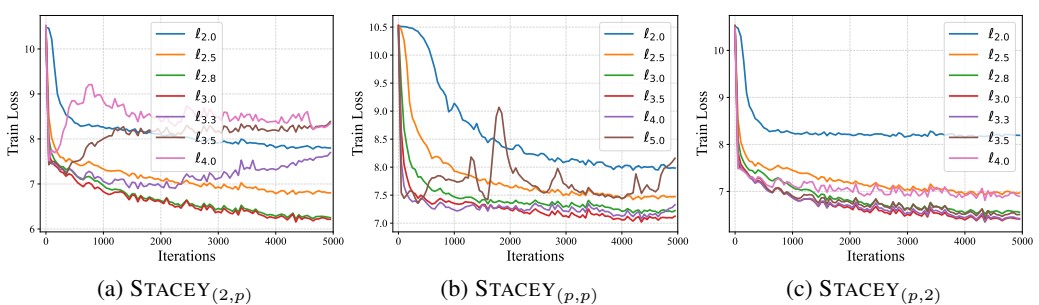

(a) STACEY$_{(2,p)}$      (b) STACEY$_{(p,p)}$      (c) STACEY$_{(p,2)}$

Figure 11: Training loss of pretraining Llama on C4 dataset with varying $\ell_p$-norm.

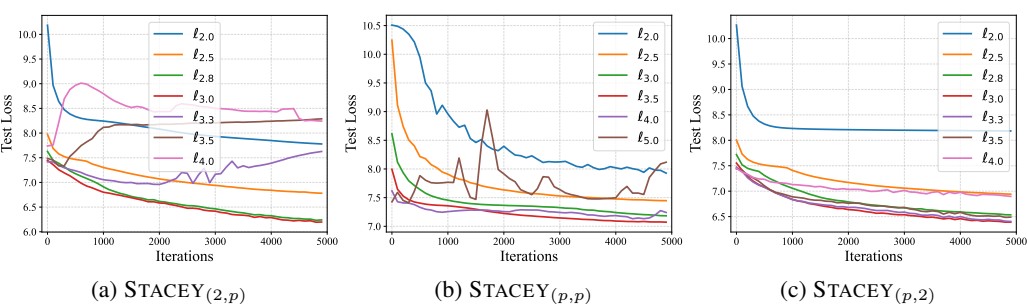

(a) STACEY$_{(2,p)}$      (b) STACEY$_{(p,p)}$      (c) STACEY$_{(p,2)}$

Figure 12: Testing loss of pretraining Llama on C4 dataset with varying $\ell_p$-norm.

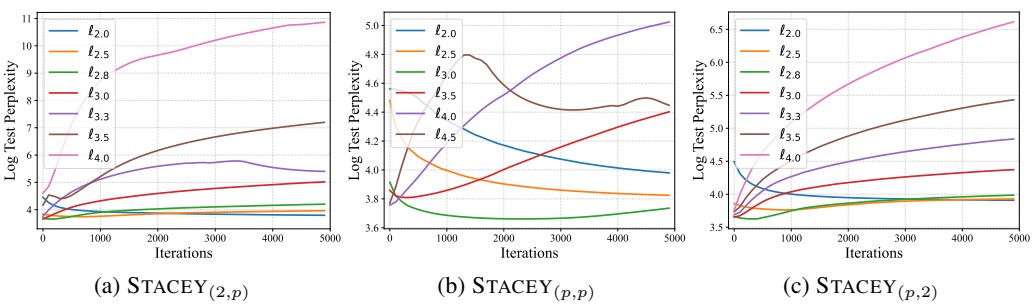

(a) STACEY$_{(2,p)}$      (b) STACEY$_{(p,p)}$      (c) STACEY$_{(p,2)}$

Figure 13: Log testing perplexity of pretraining Llama on C4 dataset with varying $\ell_p$-norm.

