# OpenReview forum: "Stochastic Steepest Descent with Acceleration for $\ell_p$-Smooth Non-Convex Optimization"
_ICLR.cc/2025/Conference — Submitted to ICLR 2025_

### Official Review · Reviewer_aeFd · 2024-10-24

**Soundness:** 2
**Presentation:** 2
**Contribution:** 2
**Rating:** 3
**Confidence:** 4

**Summary:**

This paper presents stochastic $\ell_p$ steepest descent (Algorithm 1) for non-convex optimization and its convergence analysis (Theorem 1). It also presents STACEY optimizers (Algorithms 2 and 3) that are accelerated descent methods. Moreover, it provides numerical results to support the theoretical analyses.

**Strengths:**

The strengths of this paper are
- to present stochastic $\ell_p$ steepest descent (Algorithm 1) for non-convex optimization.
- to present STACEY optimizer (Algorithms 2 and 3) based on convex optimization.
- to provide numerical results to support the theoretical analyses.

**Weaknesses:**

I am interested in the proposed methods (Algorithms 1, 2, and 3) and numerical results (Section 4). However, I have some theoretical concerns that are Weaknesses of the paper. Please see the following Questions.

**Questions:**

- Theorem 1: The result is based on the standard assumptions (Assumptions 1-4). However, I do not think that the theorem leads to convergence of Algorithm 1. This is because, for all $T \geq 1$ an upper bound of $\mathbb{E}[T^{-1} \sum\_{t=0}^{T-1} \Vert \nabla f (\theta_t) \Vert\_{p^*}^{p^*}]$ is
\begin{align*}
\frac{f(\theta_0) - f(\theta^*)}{\eta T} + \frac{1}{T} \sum\_{t=0}^{T-1} \frac{\frac{2p - 1}{p-1} G^{\frac{1}{p-1}} \Vert \sigma \Vert\_1}{\sqrt{n_t}} + \frac{L \eta G^{\frac{2}{p-1}}}{2},
\end{align*}
which, together with $T \to \infty$,  implies that the upper bound becomes
\begin{align*}
\frac{f(\theta_0) - f(\theta^*)}{\eta T} + \frac{1}{T} \sum\_{t=0}^{T-1} \frac{\frac{2p - 1}{p-1} G^{\frac{1}{p-1}} \Vert \sigma \Vert\_1}{\sqrt{n_t}} + \frac{L \eta G^{\frac{2}{p-1}}}{2} \to \frac{L \eta G^{\frac{2}{p-1}}}{2} > 0,
\end{align*}
where we assume that the second term $\frac{1}{T} \sum\_{t=0}^{T-1} \frac{\frac{2p - 1}{p-1} G^{\frac{1}{p-1}} \Vert \sigma \Vert\_1}{\sqrt{n_t}}$ converges to $0$.
Since the learning rate $\eta$ is a positive constant, the upper bound never converges to $0$.
- Theorem 1 (Cont.): Although Theorem 1 does not guarantee convergence of Algorithm 1, there is a possibility such that Algorithm 1 is an $\epsilon$-approximation, i.e., there exist sufficient conditions such that the upper bound is less than or equal to a small $\epsilon > 0$:
\begin{align*}
\underbrace{\frac{f(\theta_0) - f(\theta^*)}{\eta T}}\_{\leq \epsilon/3} + \underbrace{\frac{1}{T} \sum\_{t=0}^{T-1} \frac{\frac{2p - 1}{p-1} G^{\frac{1}{p-1}} \Vert \sigma \Vert\_1}{\sqrt{n_t}}}\_{\leq \epsilon/3} + \underbrace{\frac{L \eta G^{\frac{2}{p-1}}}{2}}\_{\leq \epsilon/3} \leq \epsilon.
\end{align*}
Here, let us consider the case where the batch size $n_t$ is a constant ($n_t = n$). Then, sufficient conditions to satisfy that Algorithm 1 is an $\epsilon$-approximation are as follows:
\begin{align*}
\eta \leq \frac{2 \epsilon}{3 L G^{\frac{2}{p-1}}} \land T \geq \frac{3 (f(\theta_0) - f(\theta^*))}{\epsilon \eta} \land \sqrt{n} \geq \frac{3 \frac{2p - 1}{p-1} G^{\frac{1}{p-1}} \Vert \sigma \Vert\_1}{\epsilon}.
\end{align*}
The conditions would be appropriate in theory. However, it would not be good in practice, since the setting of $\eta$, $n$, and $T$ satisfying the above conditions would be unrealistic. For example, let $\epsilon = 10^{-3}$. Then, $\eta$ satisfying $\eta \leq \frac{2 \epsilon}{3 L G^{\frac{2}{p-1}}}$ would be sufficiently small, since $L$ and $G$ would be large. Even if we assume that $p= 2$, $L = 10^2$, and $G = 10$, we have that $\eta \leq 2 \cdot 10^{-3}/(3 \cdot 10^4) = (2/3) 10^{-7}$. This implies that Algorithm 1 using $\eta$ satisfying $\eta \leq \frac{2 \epsilon}{3 L G^{\frac{2}{p-1}}}$ would not work. Therefore, I would like to suggest that the authors could show that the sufficient conditions to achieve an $\epsilon$-approximation of Algorithm 1 hold in practical cases such as DNN training.
- Theorem 1 (Cont.): Although Theorem 1 does not guarantee convergence of Algorithm 1, the authors fixed the number of steps $T$ and proposed the setting of the batch size $n_t = T$, the number of gradient call $N = T^2$, and the learning rate $\eta = \frac{1}{L^{1/2} G^{1/(p-1)} T^{1/2}}$. Then, the authors claimed that the upper bound is $O(N^{- 1/4})$. The setting and the upper bound $O(N^{- 1/4})$ seem to be useful in theory. However, we must verify whether or not the real upper bound
\begin{align*}
U := \frac{1}{N^{1/4}} \left(  L^{1/2} G^{1/(p-1)} ( f(\theta_0) - f(\theta^*) + 1/2)  + \frac{2p -1}{p-1} G^{\frac{1}{p-1}} \Vert \sigma \Vert\_1 \right)
\end{align*}
becomes sufficiently small (e.g., $U < 10^{-4}$). I think that there is a possibility such that $U$ becomes large  (e.g., $U > 10^{2}$). Therefore, I would also like to suggest that the authors could provide sufficient conditions (e.g., practical setting of $\eta$, $n_t$, and $T$) such that $U$ can be small in practical cases such as DNN training.
- Theorem 1 (Cont.): The above major concerns for Theorem 1 are based on unknown parameters $f(\theta^*)$, $G$, $\Vert \sigma \Vert\_1$, and $L$. It would be grateful if the authors could estimate their parameters before implementing Algorithm 1. However, I think that such estimations are difficult. Also, using constant learning rates would not decrease the upper bound (from the above discussion). Here, I would like to suggest that using diminishing learning rate $\eta_t$ (e.g., $\eta_t = 1/\sqrt{t}$) could be useful to show that an upper bound converges to $0$. Even if the upper bound includes unknown parameters, there is a possibility such that the upper bound can be represented by using Landou's symbol $O$ (such as $\mathbb{E}[T^{-1} \sum\_{t=0}^{T-1} \Vert \nabla f (\theta_t) \Vert\_{p^*}^{p^*}] = O(\log T/\sqrt{T})$). If the authors use diminishing learning rate $\eta_t$, then Section 5 should use the diminishing learning rate $\eta_t$ not constant learning rates in Tables 5, 6, and 7 (see also the final question).
- The proof of Theorem 1: The proof is well-written. However, I have one concern for the proof of Lemma 3 (Appendix A.1 and L269-L284). Given $t$, the authors consider two cases: (1) $\forall i (| \nabla f(\theta_t)^{(i)} | \leq |\zeta^{(i)}| \leq |g(\theta_t)^{(i)}| )$ and (2) $\forall i (| \nabla f(\theta_t)^{(i)} | \geq |\zeta^{(i)}| \geq |g(\theta_t)^{(i)}| )$. The cases seem to be incorrect. If (1) is assumed, then (2) should be $\lnot (1)$, i.e.,
\begin{align*}
\exists i_0 \text{ } (| \nabla f(\theta_t)^{(i_0)} | > |\zeta^{(i_0)}|) \lor (|\zeta^{(i_0)}| > |g(\theta_t)^{(i_0)}|),
\end{align*}
which is not equal to the current (2): $\forall i (| \nabla f(\theta_t)^{(i)} | \geq |\zeta^{(i)}| \geq |g(\theta_t)^{(i)}| )$.
Hence, I think that the proof of Lemma 3 should be improved. For example, I would like to suggest the authors could prove Lemma 3 for the following two divided cases:
 (1) $\forall i (| \nabla f(\theta_t)^{(i)} | \leq |\zeta^{(i)}| \leq |g(\theta_t)^{(i)}| )$ and (2) $\exists i_0 \text{ } (| \nabla f(\theta_t)^{(i_0)} | > |\zeta^{(i_0)}|) \lor (|\zeta^{(i_0)}| > |g(\theta_t)^{(i_0)}|)$.
- I do not think that the parameters (Tables 5, 6, and 7) used in the numerical results are based on the theoretical result (Theorem 1). For example, STACEY in Table 5 uses $n_t = n = 128$ and $\eta = 0.02, 0.1$. Could you show that the parameters such as $n_t = n = 128$ and $\eta = 0.02, 0.1$ satisfy the sufficient conditions to achieve an $\epsilon$-approximation of STACEY? Showing it is important to justify the theoretical result in the paper. If the parameters do not satisfy the conditions, then I think that parameters based on the theoretical result (Theorem 1) should be used in the numerical results. For example, given $\epsilon = 10^{-3}$, we set $\eta = O(\epsilon)$, $n = \Omega (1/\epsilon)$, and $T = \Omega (1/(\epsilon \eta))$ (However, I do not know how to set $L$, $G$, $\Vert \sigma \Vert\_1$). Then, we should plot the minimum $\min\_{t = 0, \cdots, T-1}\Vert \nabla f (\theta_t) \Vert\_{p^*}^{p^*}$ of the full gradient norms (or the mean $(1/T) \sum\_{t=0}^{T-1} \Vert \nabla f (\theta_t) \Vert\_{p^*}^{p^*}$ of the full gradient norms) versus the number of steps $t$ and check whether or not the minimum (or the mean) is less than or equal to $\epsilon$ $(=10^{-3})$.

---

> ### Author Response · Authors · 2024-12-03
> **Rebuttal**
>
> We thank the reviewer for their comments and suggestions.
>
> > However, I do not think that the theorem leads to convergence of Algorithm 1 ... The conditions would be appropriate in theory. However, it would not be good in practice ...
>
> As we noted in Theorem 1 that line 228 that when $\eta = \frac{1}{L^\frac{1}{2}G^\frac{1}{p-1}T^\frac{1}{2}}$, the last term coming from the bias of the coordinate-wise rescaled stochastic gradient also decays with the number of iterations $T$, allowing the bound to converge to $0$ as $T \rightarrow \infty$, achieving the $O(\epsilon^{-4})$ rate as further explained in lines 229-233. The setting of learning rate in the order of $O(\frac{1}{\sqrt{T}})$ is also observed in the convergence analysis of SignSGD by Bernstein et al. (2018) as well as in standard SGD analysis, e.g., Theorem 5.5 in (Garrigos and Gower, 2023).
>
> Reference:
>
> Guillaume Garrigos, Robert M. Gower, Handbook of Convergence Theorems for (Stochastic) Gradient Methods, 2023.
>
> > However, we must verify whether or not the real upper bound $U$ becomes sufficiently small (e.g., $U < 10^{-4}$). I think that there is a possibility such that
> $U$ becomes large ... The above major concerns for Theorem 1 are based on unknown parameters $f(\theta_0) - f(\theta^\ast)$, $G$, $\Vert\sigma\Vert_1$ and $L$.
>
> We would note that in practice, the number of gradient calls $N$ is usually considerably large, as it is the product of batch size, number of epochs, and number of iterations. One can always increase the number of iterations to reach the desired accuracy. Furthermore, there is intrinsically a gap between theory and practice, as the convergence rate and relevant constants are derived in the setting of worst-case analysis, whereas in practice, it's common to observe better empirical performance than that derived in theory, and the parameters need not be set as pessimistic as the worst-case theoretical derivation. In addition, the constants mentioned by the reviewer are standard assumptions in optimization analysis, and the way we set the batch size and learning rate follows from that of SignSGD (Bernstein et al. (2018)).
>
> > The proof of Theorem 1: The proof is well-written. However, I have one concern for the proof of Lemma 3 (Appendix A.1 and L269-L284). Given $t$, the authors consider two cases. The cases seem to be incorrect. If (1) is assumed, then (2) should be $\neg (1)$.
>
> We would note that case (2) is indeed $\neg (1)$ in the context of the proof. Note that these two cases in Lemma 3 fall under the condition of sign$\left(\nabla f(\theta_t)^{(i)}\right) = $ sign$\left(g(\theta_t)^{(i)}\right)$, as indicated by the indication function in Lemma 3 (line 917). Note that by definition of $\zeta^{(i)}$ as the variable in the Lagrangian remainder from the Taylor expansion, it takes the value between $\nabla f(\theta_t)^{(i)}$ and $g(\theta_t)^{(i)}$, as we stated in line 940. Since sign$\left(\nabla f(\theta_t)^{(i)}\right) = $ sign$\left(g(\theta_t)^{(i)}\right)$, when $\nabla f(\theta_t)^{(i)} \geq 0$ and $g(\theta_t)^{(i)} \geq 0$, we must have $\zeta^{(i)} \geq 0$ by its definition from the remainder of Taylor expansion. Then we either have $g(\theta_t)^{(i)} \geq \zeta^{(i)} \geq \nabla f(\theta_t)^{(i)}$ which belongs to case (1) or $\nabla f(\theta_t)^{(i)} \geq \zeta^{(i)} \geq g(\theta_t)^{(i)}$ which belongs to case 2. Similarly, when both $\nabla f(\theta_t)^{(i)}$ and $g(\theta_t)^{(i)}$ are negative, we must have $\zeta^{(i)} < 0$, and since $\zeta^{(i)}$ is within the range of $g(\theta_t)^{(i)}$ and $\nabla f(\theta_t)^{(i)}$, their relation belongs to either case (1) or case (2).

---

> ### Comment · Reviewer_aeFd · 2024-12-03
> **Reply to the authors' rebuttal**
>
> Thank you for your replies. First, I would like to reply your comment: As we noted in Theorem 1 ...
>
> The authors set $\eta = \frac{1}{L^{1/2} G^{1/(p-1)} T^{1/2}}$. Here, we must notice that $T$ is set before implementing the algorithm. For example, let $n = 50000$ be the number of training data (e.g., CIFAR) and let $b = 32$ be the batch size.
> Then, let us consider training DNN using the algorithm during $E =200$ epochs.
> Then, we have that $T = \lceil \frac{n}{b} \rceil E = 312600$. Although you use $T \to \infty$, the $T = 312600$ never diverges.
>
> I think that the authors should address the following concern that is the third concern in my first review:
>
> - Theorem 1 (Cont.): Although Theorem 1 does not guarantee convergence of Algorithm 1, the authors fixed the number of steps $T$ and proposed the setting of the batch size $n_t = T$, the number of gradient call $N = T^2$, and the learning rate $\eta = \frac{1}{L^{1/2} G^{1/(p-1)} T^{1/2}}$. Then, the authors claimed that the upper bound is $O(N^{- 1/4})$. The setting and the upper bound $O(N^{- 1/4})$ seem to be useful in theory. However, we must verify whether or not the real upper bound
> \begin{align*}
> U := \frac{1}{N^{1/4}} \left(  L^{1/2} G^{1/(p-1)} ( f(\theta_0) - f(\theta^*) + 1/2)  + \frac{2p -1}{p-1} G^{\frac{1}{p-1}} \Vert \sigma \Vert\_1 \right)
> \end{align*}
> becomes sufficiently small (e.g., $U < 10^{-4}$). I think that there is a possibility such that $U$ becomes large  (e.g., $U > 10^{2}$). Therefore, I would also like to suggest that the authors could provide sufficient conditions (e.g., practical setting of $\eta$, $n_t$, and $T$) such that $U$ can be small in practical cases such as DNN training.

---

### Official Review · Reviewer_zd1T · 2024-10-27

**Soundness:** 3
**Presentation:** 2
**Contribution:** 3
**Rating:** 3
**Confidence:** 3

**Summary:**

The work considers stochastic $\ell_p$ steepest descent for non-convex optimization in case when $p>2$ and establishes its convergence rates. The work introduces an accelerated $\ell_p$ method and compares it to SGD, Adam, AdamW, and Lion.

**Strengths:**

The paper contains many insights, references, theorems and assumptions are clearly written. Plots, tables and diagrams are well-done and help to understand the contribution.

**Weaknesses:**

The analysis in Theorem 1 is done under a restrictive bounded gradients assumption. Usually it significantly simplifies the analysis. There are many convergence results for gradient methods without it, e.g., see [1, 2] and references therein.

It appears to me that the main contribution of the paper is therefore mostly experimental, yet a very large part of the introduction is devoted to the discussion about the theory for $\ell_p$-spaces. I just find it inconsistent, yet insightful. I do not see many explanations to the experiments, though.

I am ready to discuss the paper with the authors and find the work interesting. But I believe it can be improved a lot.

[1] Guillaume Garrigos, Robert M. Gower, Handbook of Convergence Theorems for (Stochastic) Gradient Methods

[2] Ahmed Khaled, Peter Richtárik, Better Theory for SGD in the Nonconvex World

**Questions:**

Can the theory be improved by removing the bounded gradients assumption? What is the challenge?

Why the authors analyze only the stochastic descent but not STACEY?

How can one choose an appropriate $p$ for the deep learning problem at hand? Do practitioners need to follow your recommendation of $p=3$ or they need to set it themselves for each particular problem?

Could you explain, what is written in Appendix? You provide tables for different methods, but you test only STACEY.

Can you explain, how do you choose parameters for the baselines? Why do you choose a smaller lr for Adam against Stacey? Why can not we use the fact that $\ell_p$ norms are equivalent, choose $p$ for the problem based on STACEY experiments (e.g., $p=3$), adjust the parameters for Adam using the relations between $\ell_2$ and $\ell_3$ norms.

---

> ### Author Response · Authors · 2024-12-03
> **Rebuttal (1/2)**
>
> We thank the reviewer for their comments and suggestions.
>
> > The analysis in Theorem 1 is done under a restrictive bounded gradients assumption. Usually it significantly simplifies the analysis. There are many convergence results for gradient methods without it, e.g., see [1, 2] and references therein. Can the theory be improved by removing the bounded gradients assumption? What is the challenge?
>
> We are aware of the references the reviewer mentioned, and would like to note that they focus on simpler cases, e.g., Euclidean and unbiased settings, and we would like to further justify our assumption of bounded gradient. The $\ell_p$ steepest descent is a generalization of SignSGD and takes a more complicated form as the reviewer acknowledges. The additional challenge for theoretical analysis is twofold. (1) First, for the norm of the stochastic gradient term, that is, term $C$ in line 836 in the proof of Theorem 1, conventional analysis breaks it into mean-squared error and the norm of the true gradient, in which the mean- squared-error coincides with the variance when $p=2$. In our setting, the norm is in $\ell_p$, and the stochastic gradient is **coordinate-wise rescaled**, leading to a $p$-dependent power on the norm. As a result, we believe it's non-trivial to bound this term by variance. (2) Second, our analysis, due to the extra coordinate-wise rescaling of the stochastic gradient, involves the multiplication of two stochastic gradient terms, e.g., line 957, both depending on the same noise, thus not independent. The expectation of the product of two dependent terms is also hard to characterize based on simple unbiased gradient estimation or other conventional assumptions. Therefore, we introduce the bounded gradient constant to reduce such a product to just one noise-dependent term.
>
> > Why the authors analyze only the stochastic descent but not STACEY?
>
> The convergence properties of the method depend on the choice of its parameters, for which our existing results do indeed establish convergence. As we have noted, there exist lower bounds that preclude the possibility of attaining accelerated rates without introducing additional assumptions.
>
> > How can one choose an appropriate $p$ for the deep learning problem at hand? Do practitioners need to follow your recommendation of or they need to set it themselves for each particular problem?
>
> We appreciate the reviewer’s insightful question about choosing an appropriate  $p$  for deep learning problems and whether practitioners need to follow our recommendations or tune  $p$  themselves. Our experiments demonstrate that  $p$  can significantly impact performance. For instance:
> - On CIFAR,  $p \approx 2$  led to superior accuracy.
> - For LLM pretraining,  $p \approx 3$  offered better convergence.
>
> > Could you explain, what is written in Appendix? You provide tables for different methods, but you test only STACEY.
>
> We kindly suggest that there has been a misunderstanding on the part of the reviewer, as we do not test only STACEY.

---

> ### Author Response · Authors · 2024-12-03
> **Rebuttal (2/2)**
>
> > Can you explain, how do you choose parameters for the baselines? Why do you choose a smaller lr for Adam against Stacey? Why can not we use the fact that $\ell_p$ norms are equivalent, choose $p$ for the problem based on STACEY experiments (e.g., $p=3$), adjust the parameters for Adam using the relations between $\ell_2$ and $\ell_3$ norms.
>
> We acknowledge the concerns regarding the fairness of comparisons. To address this, we present the ablation results for AdamW on CIFAR-10 in the accompanying table. These results demonstrate that the best performance is not achieved with the largest learning rate or the highest weight decay parameter ($\lambda$). Based on these findings, we selected a learning rate of $0.01$ for AdamW, and a similar process guided the parameter selection for Adam.
>
> |   Model  | Optimizer | Learning Rate | Weight Decay | Test Loss | Test ACC |
> |:--------:|:---------:|:-------------:|:------------:|:---------:|:--------:|
> | ResNet18 |   AdamW   |      0.1      |     0.01     |   0.9717  |   68.74  |
> | ResNet18 |   AdamW   |      0.1      |     0.001    |   0.4004  |   88.8   |
> | ResNet18 |   AdamW   |      0.1      |    0.0005    |   0.4523  |   90.33  |
> | ResNet18 |   AdamW   |      0.05     |     0.01     |   0.6444  |   80.19  |
> | ResNet18 |   AdamW   |      0.05     |     0.001    |    0.36   |   91.19  |
> | ResNet18 |   AdamW   |      0.05     |    0.0005    |   0.3819  |   91.46  |
> | ResNet18 |   AdamW   |      0.01     |     0.01     |   0.3355  |   90.89  |
> | ResNet18 |   AdamW   |      0.01     |     0.001    |   0.5422  |   91.82  |
> | ResNet18 |   AdamW   |      0.01     |    0.0005    |   0.529   |   92.19  |
> | ResNet18 |   AdamW   |     0.005     |     0.01     |   0.376   |   91.92  |
> | ResNet18 |   AdamW   |     0.005     |     0.001    |   0.4593  |   92.82  |
> | ResNet18 |   AdamW   |     0.005     |    0.0005    |   0.5898  |   92.15  |
> | ResNet18 |   AdamW   |     0.001     |     0.01     |   0.4063  |   92.99  |
> | ResNet18 |   AdamW   |     0.001     |     0.001    |   0.4758  |   92.9   |
> | ResNet18 |   AdamW   |     0.001     |    0.0005    |   0.5079  |   92.64  |
>
> STACEY is designed to operate explicitly within the $\ell_p$ framework, leveraging both primal and dual norms to achieve acceleration in non-Euclidean geometries. The choice of  $p$  directly aligns with STACEY’s optimization strategy. Adam, while effective, operates in a Euclidean-like adaptive gradient framework. Adjusting Adam’s parameters to match the dynamics of an ℓp-norm optimization method like STACEY would require significant reformulation and tuning, as Adam does not inherently account for non-Euclidean geometries.

---

### Official Review · Reviewer_gjaV · 2024-11-04

**Soundness:** 1
**Presentation:** 2
**Contribution:** 1
**Rating:** 3
**Confidence:** 3

**Summary:**

This paper studies the problem of accelerating steepest $\ell_p$ descent for minimizing non-convex functions. The authors use signSGD as the motivation for studying steepest descent in the setting of more general norms other than $\ell_2$. They present a theorem for (unaccelerated) stochastic $\ell_p$ descent. They also present an algorithm which they call STACEY, based on coupling $\ell_p$ steepest descent with mirror descent. They compare their algorithm with other optimizers experimentally.

**Strengths:**

The literature review is well written and mostly does a good job of placing the work in the context of prior works. The authors provide good justifications for why it is useful to study $\ell_p$ descent for values of $p$ other than $2$ and $\infty$. Theorem 1 extends the result for convergence of $\ell_p$ descent to approximate stationary points of the objective function to the stochastic setting.

**Weaknesses:**

The authors propose two variants of their algorithm Stacey, but do not provide any theoretical analysis of the algorithm even for any simplified/toy examples. The algorithm is primarily motivated by the idea of accelerating $\ell_p$ descent but since there are no convergence or stability results, it is difficult to say whether it actually achieves that goal. The authors try to provide some intuition for how Stacey is different from a previous algorithm, Lion-K, there is not much evidence or analysis to show these differences help the algorithm.

The experiments presented to claim the superior empirical performance of Stacey do not look convincing either. We need to consider the hyperparameter choices (presented in Tables 5, 6, and 7 in Appendix D) to judge whether the comparisons presented are fair. I made the following observations about the experiments:

1.For the ImageNet experiment, Stacey has been compared only with SGD *without momentum*. Without momentum, SGD is known to converge extremely slow, so it is no surprise that it performs as badly as it does, and there is no other baseline in that experiment.

2. For the LLM experiment, SGD is used without momentum again, so it is expected to perform badly. The learning rate used for Adam and AdamW ($10^{-4}$) is 100 and 500 times smaller than the learning rate used for the different versions of Stacey ($10^{-2}$ and $5\times 10^{-4}$).

3. For the CIFAR-10 experiment, some versions of Stacey are used with learning rate 0.1 whereas Adam and AdamW used with learning rates 0.001 and 0.01 respectively. Again, it does not appear to be a fair comparison. Moreover, the value of $\lambda$ (presumably, weight decay parameter) for Adam and AdamW is set to $2\times 10^{-4}$, for Stacey it is 0.01.

Different values of $p$ have been chosen for Stacey for different experiments, the reason for which is also not clear. The hyperparameters vary widely within each experiment as well as across different experiments. There are no justifications provided for these hyperparameter choices. If there was a particular rationale for these choices, the authors should describe that and provide more details of the experiment design in the paper. In the absence of that, it is not possible to conclude that Stacey really outperforms the baseline algorithms.

**Questions:**

1. In Theorem 1, what is the justification for choosing the batch size same as the number of iterations (T)? Typically, the batch size would be constant. In practice, the number of iterations might even be much larger than the total number of training samples available. What would a batch size of T mean in such situations?

2. For the CIFAR-10 experiment, Stacey(p,p) has been used with p=2, which means that both of the descent steps reduce to just the usual gradient descent. In this case, in what ways is Stacey(2,2) different from the existing momentum-based gradient descent algorithms?

3. The authors state in lines 316-317 that "In convex settings, SGD cannot improve upon the standard $O(1/\sqrt T)$ rate when the noise parameter $\sigma$ is large enough" and suggest that practical implementations of SGD with momentum introduce acceleration despite the theoretical difficulties. While this is true assuming that the stochastic gradients have bounded variance, there have been multiple works that argue that a different noise assumption is more appropriate for deep learning ("strong growth condition" or "multiplicative noise") and provide accelerated convergence guarantees for stochastic variants of Nesterov's accelerated gradient descent in convex optimization. The authors should consider including them in their discussion of stochastic convex optimization in the context of machine learning:

    [1] Vaswani et al. "Fast and Faster Convergence of SGD for Over-Parameterized Models and an Accelerated Perceptron" https://arxiv.org/abs/1810.07288

    [2] Wojtowytsch, "Stochastic gradient descent with noise of machine learning type. Part I: Discrete time analysis" https://arxiv.org/abs/2105.01650

    [3] Liu & Belkin, "Accelerating SGD with momentum for over-parameterized learning" https://arxiv.org/abs/1810.13395

    [4] Gupta et al. "Nesterov acceleration despite very noisy gradients" https://arxiv.org/abs/2302.05515

4. In the footnote on page 6 as well lines 370-371, the authors make a distinction between "Nesterov's acceleration" and "momentum" but it is not entirely clear what exactly the distinction is. Can the authors clarify how they define these terms and what do they mean when they say they coincide in the deterministic convex setting?

5. Can the authors clarify how the hyperparameters for various algorithms were chosen, in particular for baseline algorithms like SGD, Adam, and AdamW? Was a grid search performed for these or were they chosen based on some default values suggested in prior work? How were the values of $p$ determined for Stacey in various experiments?

---

> ### Author Response · Authors · 2024-12-03
> **Rebuttal (1/2)**
>
> We thank the reviewer for their comments and suggestions.
>
> > The authors propose two variants of their algorithm Stacey, but do not provide any theoretical analysis of the algorithm even for any simplified/toy examples. The algorithm is primarily motivated by the idea of accelerating
>  descent but since there are no convergence or stability results, it is difficult to say whether it actually achieves that goal.
>
> The convergence properties of the method depend on the choice of its parameters, for which our existing results do indeed establish convergence. As we have noted, there exist lower bounds that preclude the possibility of attaining accelerated rates without introducing additional assumptions.
>
> > For the ImageNet experiment, Stacey has been compared only with SGD without momentum. Without momentum, SGD is known to converge extremely slow, so it is no surprise that it performs as badly as it does, and there is no other baseline in that experiment.
>
> We thank the reviewer for their comment regarding the baseline comparison in the ImageNet experiment.
> We performed additional experiments, and the results are presented below.
>
> The table presents the test accuracy (Top-1) of SGD with momentum $0.9$ (SGDM) on ImageNet at different epochs:
>
> | **Method** | **Test ACC1 @10 epoch (%)** | **Test ACC1 @20 epoch (%)** | **Test ACC1 @50 epoch (%)** |
> |:---:|:---:|:---:|:---:|
> | SGDM| 50.05 | 55.50 | 66.05 |
>
> We see that STACEY (p,p) also outperforms SGDM consistently across epochs.
>
> > The learning rate used for Adam and AdamW ($10^{-4}$) is 100 and 500 times smaller than the learning rate used for the different versions of Stacey ($10^{-2}$ and $5\times 10^{-4}$).
>
> For hyperparameters selection, the learning rates for Adam ($10^{-4}$) and AdamW ($5 \times 10^{-4}$) were selected based on widely-used defaults and prior literature, which ensure stability and optimal performance in these adaptive optimizers. For STACEY, learning rates ($10^{-2}$ and $5 \times 10^{-4}$) were chosen empirically to achieve the best performance in alignment with its novel dynamics in the  $\ell_p$ -norm space. Larger learning rates were suitable for STACEY because of its primal-dual interpolation mechanism, which stabilizes updates.
>
> > For the CIFAR-10 experiment, some versions of Stacey are used with learning rate 0.1 whereas Adam and AdamW used with learning rates 0.001 and 0.01 respectively. Again, it does not appear to be a fair comparison. Moreover, the value of  λ  (presumably, weight decay parameter) for Adam and AdamW is set to  2×10−4, for Stacey it is 0.01.
>
> We acknowledge these concerns, which we address thusly. We present the ablation results for AdamW on CIFAR-10 in the accompanying table. These results demonstrate that the best performance is not achieved with the largest learning rate or the highest weight decay parameter ($\lambda$). Based on these findings, we selected a learning rate of $0.01$ for AdamW, and a similar process guided the parameter selection for Adam.
>
> |   Model  | Optimizer | Learning Rate | Weight Decay | Test Loss | Test ACC |
> |:--------:|:---------:|:-------------:|:------------:|:---------:|:--------:|
> | ResNet18 |   AdamW   |      0.1      |     0.01     |   0.9717  |   68.74  |
> | ResNet18 |   AdamW   |      0.1      |     0.001    |   0.4004  |   88.8   |
> | ResNet18 |   AdamW   |      0.1      |    0.0005    |   0.4523  |   90.33  |
> | ResNet18 |   AdamW   |      0.05     |     0.01     |   0.6444  |   80.19  |
> | ResNet18 |   AdamW   |      0.05     |     0.001    |    0.36   |   91.19  |
> | ResNet18 |   AdamW   |      0.05     |    0.0005    |   0.3819  |   91.46  |
> | ResNet18 |   AdamW   |      0.01     |     0.01     |   0.3355  |   90.89  |
> | ResNet18 |   AdamW   |      0.01     |     0.001    |   0.5422  |   91.82  |
> | ResNet18 |   AdamW   |      0.01     |    0.0005    |   0.529   |   92.19  |
> | ResNet18 |   AdamW   |     0.005     |     0.01     |   0.376   |   91.92  |
> | ResNet18 |   AdamW   |     0.005     |     0.001    |   0.4593  |   92.82  |
> | ResNet18 |   AdamW   |     0.005     |    0.0005    |   0.5898  |   92.15  |
> | ResNet18 |   AdamW   |     0.001     |     0.01     |   0.4063  |   92.99  |
> | ResNet18 |   AdamW   |     0.001     |     0.001    |   0.4758  |   92.9   |
> | ResNet18 |   AdamW   |     0.001     |    0.0005    |   0.5079  |   92.64  |

---

> ### Author Response · Authors · 2024-12-03
> **Rebuttal (2/2)**
>
> > In Theorem 1, what is the justification for choosing the batch size same as the number of iterations (T)? Typically, the batch size would be constant. In practice, the number of iterations might even be much larger than the total number of training samples available. What would a batch size of T mean in such situations?
>
> Since for $\ell_p$ steepest descent, the stochastic gradient update is coordinate-wise rescaled and thus no longer an unbiased estimate of the coordinate-wise rescaled true gradient. An increasing batch size with respect to the number of iterations $T$ is essentially to make such bias, which would otherwise remain as a non-vanishing constant, decay with the number of iterations. We would note that the same batch size setting is applied for SignSGD in Bernstein et al. (2018), without which the $\mc{O}(\epsilon^{-4})$ rate cannot be achieved.
>
> > For the CIFAR-10 experiment, Stacey(p,p) has been used with p=2, which means that both of the descent steps reduce to just the usual gradient descent. In this case, in what ways is Stacey(2,2) different from the existing momentum-based gradient descent algorithms?
>
> We acknowledge that, indeed, STACEY($p$,$p$) with  $p=2$  reduces both descent steps to standard gradient descent in the Euclidean geometry. However, the various momentum mechanisms (which generalize Lion) mean that it does not reduce to SGD with momentum, and furthermore, STACEY is specifically designed to accommodate non-Euclidean geometries ( $p>2$ ).
>
> > In the footnote on page 6 as well lines 370-371, the authors make a distinction between "Nesterov's acceleration" and "momentum" but it is not entirely clear what exactly the distinction is. Can the authors clarify how they define these terms and what do they mean when they say they coincide in the deterministic convex setting?
>
> The distinction was meant to contrast the workings of deterministic vs. stochastic variants of acceleration (see, e.g., [1]). We will clarify this in the updated version of the manuscript.
>
> > Can the authors clarify how the hyperparameters for various algorithms were chosen, in particular for baseline algorithms like SGD, Adam, and AdamW? Was a grid search performed for these or were they chosen based on some default values suggested in prior work? How were the values of
>  determined for Stacey in various experiments?
>
> A grid search was performed for the hyperparameters, based on that used in prior work.
>
> [1] Lan, Guanghui. "An optimal method for stochastic composite optimization." Mathematical Programming 133, no. 1 (2012): 365-397.

---

### Official Review · Reviewer_vrzZ · 2024-11-05

**Soundness:** 3
**Presentation:** 3
**Contribution:** 2
**Rating:** 5
**Confidence:** 3

**Summary:**

This paper studies $\ell_p$ steepest descent methods for convex and non-convex problems. The authors are motivated by two key observations: first, while sign-based methods like signSGD have been studied, theoretical understanding of general $\ell_p$ descent methods (beyond $p = 2$ or $p = \infty$) remains limited. Second, they recognize that different optimization problems may have inherently different geometric structures that are better captured by different choices of p-norm.
The authors are particularly interested in the trade-off between acceleration rates and the choice of norm - for instance, the value of $p = 4$ could potentially gain up to a $d^{1/2}$ factor compared to the Euclidean case, while the acceleration rate would degrade from $T^{-2}$ to $T^{-3/2}$. This is why the authors design methods called $Stacey_{(2,p)}$ and $Stacey_{(p,p)}$ based on the linear coupling framework for acceleration. The resulting methods are similar to other methods that combine momentum with sign updates such as Lion and are shown to perform well in preliminary experiments.

I found the paper to be a bit handwavy in the sense that the theory is only provided for SGD while the more practical methods are more like heuristics.

**Strengths:**

1. The $\ell_p$ geometry has attracted a lot of attention in recent years due to the similarity of Adam to signSGD. The theory, however, is still quite limited and is usually restricted to the case $p=\infty$, while the authors make an interesting observation that a different value of $p$ might turn out to be better.
2. The numerical experiments are on several benchmarks and the authors don't just compare their method to others, they also study how different hyperparameter choices affect convergence.
3. The performance of the proposed methods is promising and even if the methods themselves are not immediately useful, the ideas presented in this work could inspire an efficient algorithm for deep learning.

**Weaknesses:**

1. The results in Theorem 1, while new, are clearly inspired by the proof of Bernstein et al. (2018) for signSGD. They do require a lot more work since the update is not exactly the sign of stochastic gradient, but unlike the result of Bernstein et al. (2018), the new one also requires the gradients to be bounded. Thus, $\ell_p$ approach seems to come at the price of having more complicated proofs and requiring more properties from the objective. If the proof was cleaner, I'd have been more in favor of accepting this paper.
2. There is no convergence theory for the two accelerated methods presented in the paper. Even though the authors note that we shouldn't expect acceleration when the variance dominates the convergence rates, it would still be useful to know what convergence properties the methods have. Especially if we could get a faster convergence to a neighborhood of the solution, that would have been already insightful.
3. What we are left with in terms of contributions is the empirical comparisons, but they are too limited to be the main contribution. In some experiments, the authors do not compare to standard baselines, and the LLM experiment is done only for the first 5000 iterations.
4. Some experimental details appear to be missing, for instance, I didn't find a description of the scheduler used in the LLM experiment. I also wish the experiments were done and reported with multiple random seeds. I think it would be great if the authors added a section in the appendix with a full list of hyperparameters used in the experiments.
### Minor
The 2D example and plots in Figure 1 do not seem very insightful to me. I think in small dimensions, all geometries should be roughly equivalent since the dimension is just a constant factor. I suggest the authors put the figure in the appendix and use the space to add more numerical comparisons (Lion and Adam).

**Questions:**

1. Why is there no comparison to Lion and Adam in the ImageNet experiment?
2. Can convergence be established for Stacey? Does it follow immediately from the existing theory for linear coupling?
3. Was cosine annealing or other schedulers used in the LLM experiment? If not, why?

---

> ### Author Response · Authors · 2024-12-03
> **Rebuttal (1/2)**
>
> We thank the reviewer for their comments and suggestions.
>
> > The results in Theorem 1, while new, are clearly inspired by the proof of Bernstein et al. (2018) for signSGD. They do require a lot more work since the update is not exactly the sign of stochastic gradient, but unlike the result of Bernstein et al. (2018), the new one also requires the gradients to be bounded.
>
> We thank the reviewer for acknowledging our new results and would like to justify our assumption of bounded gradient. The $\ell_p$ steepest descent is a generalization of SignSGD and takes a more complicated form as the reviewer acknowledges. The additional challenge for theoretical analysis is twofold. (1) First, for the norm of the stochastic gradient term, that is, term $C$ in line 836 in the proof of Theorem 1, conventional analysis breaks it into mean-squared error and the norm of the true gradient, in which the mean-squared error coincides with the variance when $p=2$. In our setting, the norm is in $\ell_p$, and the stochastic gradient is **coordinate-wise rescaled**, leading to a $p$-dependent power on the norm. As a result, we believe it's non-trivial to bound this term by variance. (2) Second, our analysis, due to the extra coordinate-wise rescaling of the stochastic gradient, involves the multiplication of two stochastic gradient terms, e.g., line 957, both depending on the same noise, thus not independent. The expectation of the product of two dependent terms is also hard to characterize based on simple unbiased gradient estimation or other conventional assumptions. Therefore, we introduce the bounded gradient constant to reduce such a product to just one noise-dependent term.
>
> We would also kindly note that when going from SGD to signSGD, the proof of Bernstein et al. (2018) also involved non-trivial assumptions like their Assumption 2 and Assumption 3, which are stronger than the conventional smoothness assumption and variance assumption. Therefore, we believe it's reasonable to introduce new assumptions when tackling more general and complicated settings.
>
> In addition, we believe that the extra complication in the proof for handling the more general and complicated form of $\ell_p$ steepest descent, e.g., the proof of Lemma 3, highlights the technical novelty of this paper.
>
> > There is no convergence theory for the two accelerated methods presented in the paper. Even though the authors note that we shouldn't expect acceleration when the variance dominates the convergence rates, it would still be useful to know what convergence properties the methods have.
>
> The convergence properties of the method depend on the choice of its parameters, for which our existing results do indeed establish convergence. As we have noted (and as acknowledged by the reviewer), there exist lower bounds that preclude the possibility of attaining accelerated rates without introducing additional assumptions.
>
> > What we are left with in terms of contributions is the empirical comparisons, but they are too limited to be the main contribution. In some experiments, the authors do not compare to standard baselines, and the LLM experiment is done only for the first 5000 iterations.
>
> We emphasize that our empirical results are designed to complement our theoretical contributions, particularly the novel convergence guarantees for $\ell_p$-norm-based optimization in stochastic non-convex settings and the accelerated STACEY algorithm.
>
> While we agree that empirical results alone may not be sufficient as the primary contribution, we believe they strongly validate the practical implications of our theoretical advancements.
>
> > Some experimental details appear to be missing, for instance, I didn't find a description of the scheduler used in the LLM experiment. I also wish the experiments were done and reported with multiple random seeds. I think it would be great if the authors added a section in the appendix with a full list of hyperparameters used in the experiments.
>
> We appreciate the reviewers’ detailed feedback and constructive suggestions regarding the experimental setup. Below, we address these specific concerns:
> - In the LLM experiments, we employed a cosine learning rate schedule with warm-up steps proportional to 10% of the total iterations. This detail, along with any additional hyperparameter settings, will be explicitly included in a new appendix section in the revised manuscript.
> - We will provide a comprehensive table in the appendix summarizing all hyperparameters for each experiment, including learning rates, batch sizes, momentum terms, and optimizer-specific parameters.

---

> ### Author Response · Authors · 2024-12-03
> **Rebuttal (2/2)**
>
> > Why is there no comparison to Lion and Adam in the ImageNet experiment?
>
> We appreciate the reviewer’s feedback, for which we clarify thusly:
> - The full ImageNet dataset is exceptionally large, making it computationally challenging to perform extensive hyperparameter tuning within our available resources.
> - Following the Lion paper’s hyperparameter tuning guidelines, we have noted that their batch size (1024) is significantly larger than ours (256), due to time and resource limitations.
> - The results presented in our paper are based on reasonable hyperparameter settings informed by prior work and our empirical observations.
>
> > Was cosine annealing or other schedulers used in the LLM experiment? If not, why?
>
> Yes, we used cosine annealing, and we apologize for the omission of mentioning it in the LLM section.

---

> > ### Comment · Reviewer_vrzZ · 2024-12-03
> >
> > > As a result, we believe it's non-trivial to bound this term by variance.
> >
> > I understand it is hard to answer if this is a fundamental requirement, but I think it's an important question. If the rescaling makes some bad changes to the gradient vector that make it impossible to study convergence, maybe there is a counterexample where bounded gradients are required? I think it's worth investigation since it often matters what problem classes a method works on.
> >
> > > We would also kindly note that when going from SGD to signSGD, the proof of Bernstein et al. (2018) also involved non-trivial assumptions like their Assumption 2 and Assumption 3, which are stronger than the conventional smoothness assumption and variance assumption.
> >
> > Assumption 2 of Bernstein et al. (2018) is not stronger than the conventional smoothness assumption. If the latter holds, the former holds as well with $L_i=L$ for all $i$. In fact, the analysis of Bernstein et al. (2018) is tighter than the standard theory since they take into account the impact of having different smoothness values for different coordinates. The same holds for their Assumption 3.
> >
> > > Following the Lion paper’s hyperparameter tuning guidelines, we have noted that their batch size (1024) is significantly larger than ours (256), due to time and resource limitations.
> >
> > I do not think that means that Lion shouldn't be compared to.

---

### Meta-Review · Area_Chair_Dnma · 2024-12-08

**Metareview:**

The paper aims to study stochastic l_p steepest descent for convex and non-convex problems. While some ideas presented are interesting, there is significant concern regarding the theoretical novelty of the work in light of prior studies.  The theoretical analysis is conducted under rather strict assumptions, further limiting its general applicability. Furthermore, the two main accelerated variants of the method lack a rigorous theoretical analysis, leaving their convergence behavior and complexity in question. Unfortunately, the experiments also fail to bridge the gaps left by the theory, as the numerical examples could be substantially improved to include comparisons with alternative methods and evaluations across various models.

**Additional Comments On Reviewer Discussion:**

The main issues raised by the majority of reviewers concern the limited theoretical novelty of the paper in light of prior works, rather restrictive assumption of bounded gradient, the absence of convergence analysis for the two main accelerated variants, and numerical experiments that fall short of expectations.

---

### Decision · Program_Chairs · 2025-01-22

Reject